# Regional variability of aerosol impacts on clouds and radiation in global kilometer-scale simulations

Ross J. Herbert[1,2], Andrew I. L. Williams[1,3], Philipp Weiss[1], Duncan Watson-Parris[1,4], Elisabeth Dingley[1,5], Daniel Klocke[6], and Philip Stier[1]

[1]Atmospheric Oceanic and Planetary Physics, Department of Physics, University of Oxford, Oxford, UK
[2]Now at: Institute for Climate and Atmospheric Science, University of Leeds, Leeds, UK
[3]Now at: Program in Atmospheric and Oceanic Science, Princeton University
[4]Now at: Scripps Institution of Oceanography and Halıcıoğlu Data Science Institute, University of California San Diego
[5]Now at: WCRP CMIP International Project Office
[6]Max Planck Institute for Meteorology, Hamburg, Germany

**Correspondence:** Ross J. Herbert (R.J.Herbert@leeds.ac.uk)

**Abstract.** Anthropogenic aerosols are a primary source of uncertainty in future climate projections. Changes to aerosol concentrations modify cloud radiative properties, radiative fluxes, and precipitation from the micro- to the global scale. Due to computational constraints, we have been unable to explicitly simulate cloud dynamics in global-scale simulations, leaving key processes, such as convective updrafts, parameterized. This has significantly limited our understanding of aerosol impacts on convective clouds and climate. However, new state-of-the-art climate models are capable of representing these scales. In this study, we used the kilometer-scale Icosahedral Nonhydrostatic (ICON) earth system model to explore the global-scale rapid response of clouds and precipitation to an idealized distribution of anthropogenic aerosol via aerosol-cloud interactions (ACI) and aerosol-radiation interactions (ARI). In our simulations over 30 days, we find that the aerosol impacts on clouds and precipitation exhibit strong regional dependence. The impact of ARI and ACI on clouds in isolation shows some consistent behavior, but the magnitude and additive nature of the effects are regionally dependent. Some regions are dominated by either ACI or ARI, whereas others behaved nonlinearly. This suggests that the findings of isolated case studies from regional simulations may not be globally representative; ARI and ACI cannot be considered independently and should both be interactively represented in modelling studies. We also observe pronounced diurnal cycles in the rapid response of cloud microphysical and radiative properties, which suggests the usefulness of using polar-orbiting satellites to quantify ACI and ARI may be more limited than presently assumed. The simulations highlight some limitations that need to be considered in future studies. Isolating kilometer-scale aerosol responses from internal variability will require longer averaging periods or ensemble simulations. It would also be beneficial to use interactive aerosols and assess the sensitivity of the conclusions to the cloud microphysics scheme.

## 1 Introduction

Aerosols and their impact on Earth's climate remain a key uncertainty for anthropogenic climate change. On a global scale, they act primarily to cool the climate, partially compensating for warming induced by greenhouse gases (Bellouin et al., 2020; Forster et al., 2021; Watson-Parris and Smith, 2022). On the regional scale, they influence clouds, precipitation, and fluxes

of radiation throughout the atmosphere. However, the magnitude of their global and regional impact remains uncertain (Gliß et al., 2021; Myhre et al., 2018; Sand et al., 2021; Williams et al., 2022).

Aerosols modify the atmosphere via two pathways: aerosol-cloud interactions (ACI) and aerosol-radiation interactions (ARI). ACI considers the role that aerosols play in their ability to act as cloud condensation nuclei or ice-nucleating particles, thus directly influencing the distribution of cloud droplets or ice particles, and modifying the radiative properties of clouds and precipitation processes. ARI considers the impact that aerosols have via their radiative properties on scattering and absorption, thereby modifying the fluxes of radiation at the surface and top-of-atmosphere (TOA) and the vertical heating profile. Both ACI and ARI can interact with cloud dynamics, leading to circulation and precipitation changes. Quantifying aerosol effects on global and regional climate is challenging due to the microphysical scales on which ACI and ARI processes fundamentally act upon, which cannot be explicitly represented in global models, limiting our ability to accurately quantify their role in the present and future climates.

An important source of uncertainty arises in the inability for models to sufficiently represent the turbulent motions that drive the vertical transport of energy and water, which has important implications for the formation and evolution of shallow and deep-convective clouds, their diurnal cycle, and interactions between aerosols, the cloud-scale, and the large-scale environment. Current Earth System Models (ESMs) use horizontal resolutions typically ranging from tens to hundreds of kilometers. On these scales, fundamental climate processes, such as mesoscale convective systems or ocean eddies remain unresolved and need to be parameterized. This requires the use of convection parameterizations, introducing significant uncertainties due to structural limitations such as locality (Wang et al., 2022), lack of convective memory (Colin et al., 2019; Tan et al., 2018), or the inability to accurately represent convective organisation and mesoscale convective systems (Mapes and Neale, 2011; Shamekh et al., 2023). ESMs also generally have very simplified representations of convective microphysics, as they do not explicitly represent vertical motions and associated cooling rates.

A wide range of aerosol effects on convective clouds have been proposed that cannot be represented in the highly parameterized configurations of current ESMs. Regional high-resolution models provide useful process insights (Marinescu et al., 2021), but their global representativeness remains unclear as the role of aerosols in one location may not be applicable to other regions (Williams et al., 2023). Regional simulations may also not adequately represent the interaction between the large-scale thermodynamic environment and the regional scale (Dagan et al., 2022). Previous work using limited area simulations with kilometer-scale resolution has shown that aerosols have the potential to significantly modify the diurnal cycle of convection and cloud evolution over widespread regions (Herbert et al., 2021a; Liu et al., 2020; Hodzic and Duvel, 2018). This is supported by observations showing aerosol perturbations can significantly modify widespread properties of clouds through changes to the development and evolution of convection (Herbert and Stier, 2023; Jiang et al., 2018; Koren et al., 2004; Yu et al., 2007).

A new generation of global kilometer-scale models is now being developed to run on scales that explicitly simulate convection – a significant step towards a more realistic representation of the Earth system (Palmer and Stevens, 2019). The DYnamics of the Atmospheric general circulation Modeled On Non-hydrostatic Domains (DYAMOND) initiative (Stevens et al., 2019) has brought together a number of these next generation models to explore their capabilities and has demonstrated that many dynamical features in the Earth System are better reproduced with resolved convection. As such, this has greatly improved the

realism (Ban et al., 2021; Kendon et al., 2019) and predictive skill (Weber and Mass, 2019) of regional precipitation magnitudes and timings across the tropics and midlatitudes. It has also been shown to improve the representation of global scale features such as the Madden-Julian Oscillation (Savarin and Chen, 2022), demonstrating the benefits of employing these models when studying global-scale teleconnections and patterns.

Although much focus has been on convective processes and associated precipitation, the role of aerosols in these new configurations remains currently poorly understood. Many of the new generation modelling frameworks include some representation of aerosol, though their role in the climate system have only been touched upon. Sato et al. (2018), for example, studied the warm-topped cloud liquid water path (LWP) response to perturbations of the aerosol optical depth (AOD) in the Non-hydrostatic Icosahedral Atmospheric Model (NICAM) model with 14 km horizontal resolution. The authors found that using an explicit representation of cloud microphysics on a global scale produced a negative LWP-AOD relationship, in agreement with satellite observations, that was not replicated in a coarser global model. The study demonstrates that ACI effects on a global-scale are sensitive to the representation of cloud processes, but did not extend the analysis to other cloud types, nor consider ARI effects. It is well established that ARI can impact convective processes over land (Andreae et al., 2004; Bukowski and van den Heever, 2021; Herbert et al., 2021a; Hodzic and Duvel, 2018; Jiang et al., 2018; Koren et al., 2008; Park and van den Heever, 2022) and ocean (Gordon et al., 2018; Williams et al., 2022). Therefore, it is important to understand its role alongside the improved representation of convection in these new-generation models.

Aerosols themselves are also a source of uncertainty in ESMs and high-resolution simulations due to complex aerosol microphysical processes that are poorly constrained or inadequately represented (White et al., 2017; Sand et al., 2021; Vogel et al., 2022; Regayre et al., 2018; Gliß et al., 2021). This complexity can also inhibit the interpretability of model behavior (Proske et al., 2023) and may not necessarily scale with improved model representation (Ekman, 2014). Previous studies have used idealized or simplified aerosol representations to remove this uncertainty and focus on quantifying aerosol interactions at the process level. Prescribed aerosol fields have been used to systematically quantify the sensitivity of the atmosphere to aerosol properties, including horizontal gradients (Lee et al., 2014), vertical profiles (Herbert et al., 2020; Johnson et al., 2004), concentrations (Dagan and Eytan, 2024; Tang et al., 2024), and spatial distributions (Williams et al., 2022; Dagan et al., 2021; Fiedler et al., 2017; Herbert et al., 2021a; Fiedler and Putrasahan, 2021). Idealized aerosol representations have also proven useful for identifying model structural uncertainties and estimating aerosol radiative forcing in intercomparison studies (Stier et al., 2013; Fiedler et al., 2019; Randles et al., 2013; Fiedler et al., 2023) and have been combined with reduced complexity climate models to provide a means of assessing sensitivity to future aerosol scenarios (Herbert et al., 2021b; Stjern et al., 2024; Recchia and Lucarini, 2023).

The emergence of next-generation kilometer-scale ESMs provides a unique opportunity to study aerosol-convection interactions and the interactions with the large-scale environment. However, at least initially, the uncertainty in explicitly simulated aerosols will remain significant, making it difficult to disentangle the complex cloud response from differences in the aerosol representation. Therefore, in this study, we examine the impact of idealized anthropogenic aerosol perturbations on the climate using global storm-resolving simulations with the ICOsahedral Nonhydrostatic (ICON) model (Hohenegger et al., 2023) coupled to the simple plume implementation of the Max Planck Institute Aerosol Climatology version 2 (MACv2-SP) (Stevens

et al., 2017). We analyze the rapid response of clouds and the thermodynamic environment to an aerosol perturbation by contrasting simulations using aerosol representative of the pre-industrial era with aerosol representative of the present day.

## 2 Methodology

### 2.1 Model description and setup

We use the ICON model in its Sapphire configuration, which is designed for kilometer-scale simulations of the Earth system. A detailed description and evaluation are presented by Hohenegger et al. (2020, 2023) and only briefly described here. The atmosphere is solved with the non-hydrostatic model from Zängl et al. (2015) and land is represented with the Jena Scheme for Biosphere Atmosphere Coupling in Hamburg (JSBACH) dynamic vegetation model (Reick et al., 2013). We run the model in an atmosphere-only mode, with sea surface properties (sea surface temperature and sea ice concentration) prescribed as atmospheric boundary conditions following the atmospheric model intercomparison project AMIP (Taylor et al., 2012). The atmosphere is modeled with non-hydrostatic equations for the conservation of mass, momentum, and energy as well as parameterization schemes for the unresolved physical processes. The equations are discretized on an icosahedral-based mesh and integrated with a two-level predictor-corrector scheme.

ICON includes parameterization schemes for radiation (Pincus et al., 2019), cloud microphysics (Baldauf et al., 2011), and turbulence (Smagorinsky, 1963). Turbulence is parameterized with the Smagorinsky scheme even though turbulent eddies are partially resolved at the kilometer-scale (Dipankar et al., 2015; Hohenegger et al., 2023; Smagorinsky, 1963). Radiation is parameterized with a radiative transfer scheme from Pincus et al. (2019). The scheme computes radiative properties and radiative fluxes over 14 shortwave bands and 16 longwave bands. The optical properties of clouds are sensitive to the cloud droplet number concentration, $N_\mathrm{d}$, which follows a predefined vertical profile and is discussed further in Sect. 2.2.1. Cloud microphysics are parameterized with the one-moment scheme from Baldauf et al. (2011). The scheme computes the masses of six hydrometeor classes: water vapour, cloud water, cloud ice, rain, snow, and graupel. The classes interact based on parameterized processes including condensation and autoconversion of cloud droplets to rain, the latter of which follows the description $q_\mathrm{aut} \sim N_\mathrm{d}^{-2}$ (Seifert and Beheng, 2006).

In our simulations, we use a horizontal resolution of approximately 5 km with 90 levels from the surface to 75 km corresponding to a vertical resolution of about 25 to 400 m (Hohenegger et al., 2023, G_AO_5km setting). This configuration of ICON does not explicitly resolve the smallest scales of convection (< 5 km) but has been shown to reproduce many features of the climate system relevant for this study (Hohenegger et al., 2023), including seasonal cycles of precipitation and soil moisture, the structure of the atmosphere in deep convective regions, and coupling between sea surface temperature and precipitation. Segura et al. (2022) also demonstrate that this configuration reproduces the observed diurnal cycle of tropical precipitation. Given that ESMs tend to use spatial resolutions of tens to hundreds of kilometers, this makes a marked improvement in our ability to resolve many aspects of convection (Done et al., 2004; Prein et al., 2013) and is well suited for our study. The simulations are initialized using the ERA5 meteorological reanalysis and run for a 40-day period, similar to the DYAMOND protocol (Stevens et al., 2019), which includes a 10-day period of spin-up. The prescribed oceanic properties are fixed at mean September values

for the year 2016. The month of September is chosen due to the pronounced biomass burning activity that occurs in this month around the world (van der Werf et al., 2017). This provides us with a large global mean aerosol perturbation. The use of fixed, monthly mean sea surface temperatures and sea ice reduces the noise due to atmosphere-ocean coupling and allows us to focus on the rapid response of the atmosphere and climate to the aerosol perturbation, without the confounding effects of sea surface temperature changes. Aerosol perturbations, described in the following section, are held at mean September values for the year 2016 to produce a consistent aerosol perturbation throughout the simulations.

## 2.2 Aerosol representation

In this study, natural aerosols are represented by the Max Planck Institute Aerosol Climatology version 2 (MACv2.0), described by Kinne (2019), which we will refer to as K19, and anthropogenic aerosols are represented using the simple plume implementation of MACv2.0, named MACv2-SP (Stevens et al., 2017). The K19 climatology and MACv2-SP are used in ICON to represent aerosols in the radiation scheme. We extend MACv2-SP to the cloud microphysics scheme to link the anthropogenic aerosol perturbation to the warm-rain process (auto-conversion). The prescribed fields of aerosol are non-interactive, but magnitudes are spatially and temporally variable. This is a simplified representation, but provides a means to robustly isolate the role of aerosols in the climate system (Fiedler et al., 2019, 2017) without the added complexity of aerosol microphysical processes, which are themselves an important source of uncertainty in ESMs and high-resolution simulations (White et al., 2017; Sand et al., 2021; Vogel et al., 2022; Regayre et al., 2018; Mann et al., 2014; Gliß et al., 2021).

### 2.2.1 Aerosol-radiation interactions

ARI effects are included in the ICON radiation scheme. 3D fields of aerosol extinction from natural sources in the pre-industrial era (year 1850) are taken from the K19 aerosol climatology described by Kinne (2019). Anthropogenic aerosol perturbations are represented using MACv2-SP, described in full by Stevens et al. (2017), which provides the model with 3D fields of aerosol extinction that are calculated for nine predefined plumes of aerosol concentrations and optical properties. The plumes are spatially consistent with the dominant sources of global anthropogenic aerosol emissions, and each is characterized by parameters that control its horizontal and vertical distribution, aerosol concentration and optical properties, annual cycle, and year-to-year variations. The plumes extend from the surface to the top of the model atmosphere and generally peak between 2 and 5 km. Each plume is representative of either industrial or biomass burning emissions, defined by the single-scattering albedo (0.93 or 0.87 at 500 nm) applied to the aerosol field. The plume aerosol concentrations are scaled year-to-year between 1850 and 2016, starting from 0.0 in 1850, to match the historical period. The contributions from the natural aerosol (K19) and anthropogenic aerosol (MACv2-SP) are summed to produce the prescribed fields of aerosol extinction in the ICON radiation scheme.

In our configuration of MACv2-SP, we adjust the biomass burning plumes (North Africa, South America, Southeast Asia, and South Central Africa). In the standard MACv2-SP setup for the present-day climate, anthropogenic sources account for around 40 % of the plume extinction. These figures are uncertain (Hamilton et al., 2018) and may substantially underestimate the anthropogenic contribution (Lauk and Erb, 2009). In our simulations, we enhance the anthropogenic contribution in MACv2-

SP by a factor of 1.5, which increases the anthropogenic contribution to around 50 %. This is consistent with higher estimates (Lauk and Erb, 2009, and references therein) and should provide a stronger signal in response to our perturbations. As we show in Fig. 1, the resulting distribution and magnitude of present-day AOD is consistent with observations.

### 2.2.2 Aerosol-cloud interactions

ACI effects are included in the ICON radiation scheme (cloud optical properties) and cloud microphysics scheme (autoconversion rate) using global distributions of AOD perturbations provided by MACv2-SP. The two schemes are not coupled and employ different assumptions about the cloud droplet number concentration ($N_d$). Therefore we use the variable names $N_{d,rad}$ and $N_{d,cld}$ to distinguish between the treatment of $N_d$ in the two schemes.

In the radiation scheme, the vertical profile of cloud droplet effective radius is dependent on the cloud water content, the cloud droplet number concentration $N_{d,rad}$, and a scaling factor that accounts for the width of the droplet distribution (Stevens et al., 2013, Eq. 7). $N_{d,rad}$ follows a predefined profile in the radiation scheme

$$N_{d,rad}(p) = \begin{cases} N_{d,rad-top} + (N_{d,rad-sfc} - N_{d,rad-top})\exp\left(1 - (p/800\,hPa)^2\right), & p < 800 \text{ hPa} \\ N_{d,rad-sfc}, & \text{else} \end{cases} \tag{1}$$

where $N_{d,rad-top}$ and $N_{d,rad-sfc}$ are the number concentration at the top and bottom of the atmosphere and $p$ is the pressure. In the default ICON configuration $N_{d,rad-top}$ is set to 20 cm$^{-3}$ and $N_{d,rad-sfc}$ is set to 120 cm$^{-3}$ over land and 80 cm$^{-3}$ over oceans. These are the values we use for the pre-industrial climate. For ACI effects in the present-day climate we use a spatially dependent ACI scaling factor, $f_N$, that modifies the global distribution of $N_{d,rad-sfc}$ ($N_{d,rad-top}$ is kept constant at 20 cm$^{-3}$). Following the approach of (Stevens et al., 2017, Eq. 15), $f_N$ is calculated in MACv2-SP using

$$f_N = \frac{\ln(b_N \tau_{PD} + 1)}{\ln(b_N \tau_{PI} + 1)}, \tag{2}$$

where $\tau_{PI}$ and $\tau_{PD}$ are the AOD in the pre-industrial and present-day climates and $b_N$ is a predefined parameter that describes the sensitivity of $N_d$ to AOD ($N_{d,cld} = a_N \ln(b_N \tau + 1)$, Stevens et al. (2017)). In the default MACv2-SP setup, $a_N$ and $b_N$ have values of 60 cm$^{-3}$ and 20. This provides a relatively weak sensitivity ($N_{d,cld} = 140$ cm$^{-1}$ for $\tau = 0.5$), which may be inconsistent with observations over land (Hudson and Yum, 2001; McCoy et al., 2018; Miles et al., 2000; Squires, 1958) and in the presence of convective updrafts (Braga et al., 2021; Gryspeerdt et al., 2023; Machado et al., 2018; Pringle et al., 2009) showing concentrations in excess of 300 cm$^{-3}$. In this study, we set the values of $a_N$ and $b_N$ to 410 cm$^{-3}$ and 5, taken from Herbert et al. (2021a). This provides more sensitivity than the original, but as we show in Fig. 1 results in a present day distribution of $N_d$ consistent with observations.

In the default ICON setup, the microphysics scheme uses a predefined value for the cloud droplet number concentration ($N_{d,cld}$) that is spatially invariable and constant in altitude. We use this for our PI distribution of $N_{d,cld}$, which we set to 80 cm$^{-3}$. We represent ACI effects in the microphysics scheme using the ACI scaling factor $f_N$, as calculated above. Applying $f_N$ to the pre-industrial distribution of $N_{d,cld}$ provides an idealized present-day distribution that is spatially consistent with the anthropogenic contributions in the MACv2-SP plumes.

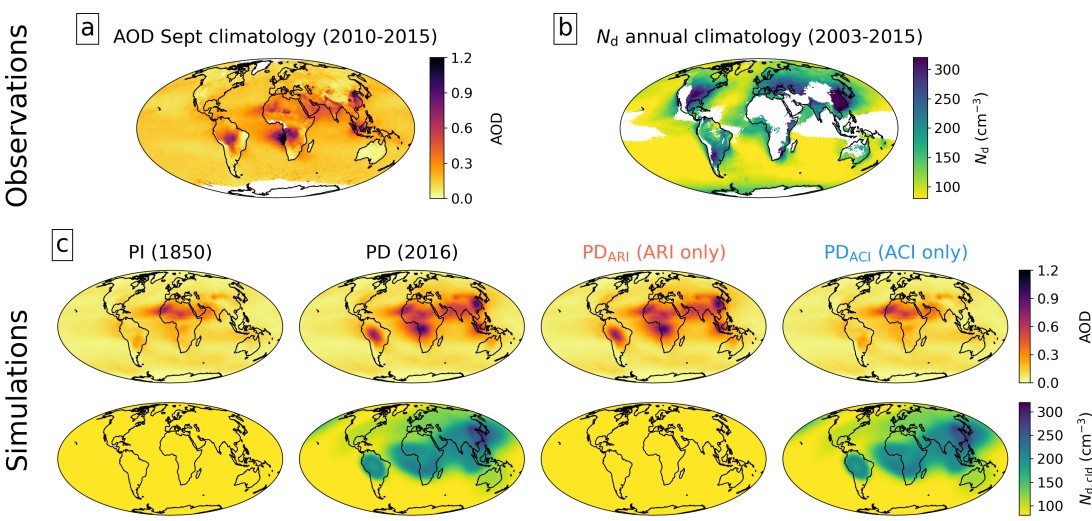

**Figure 1.** Climatologies of satellite retrieved (**a**) September mean $AOD_{550nm}$ (for 2010 – 2015) measured by Platnick et al. (2015) and (**b**) annual mean $N_d$ for cloud tops < 3.2 km (2003 – 2015; only showing grid points with 50 successful retrievals) estimated by Grosvenor et al. (2018). Panel (**c**) shows the simulated AOD from the K19 climatology and plume model MACv2-SP (top row) and $N_{d,cld}$ (bottom row) in each of the 4 simulations.

## 2.3 Simulations

We use four simulations to explore the rapid response of clouds and climate to our idealized aerosol perturbations (outlined in Table 1). The control simulation (PI) uses values that are representative of a pre-industrial atmosphere consisting of natural aerosol and background ARI and ACI effects. Global fields of natural aerosol extinction are represented by the K19 climatology for the year 1850. $N_{d,cld}$ is held constant at a value of 80 cm$^{-3}$, whilst $N_{d,rad}$ follows a vertical profile according to Eq. 1 and varies spatially with $N_{d,rad-sfc}$ set to 120 cm$^{-3}$ on land and 80 cm$^{-3}$ over oceans. A second simulation (PD) is run with values that are representative of a present-day atmosphere that includes ACI and ARI effects due to anthropogenic activity. Aerosol extinction fields from anthropogenic aerosol are represented by the plume model MACv2-SP for the year 2016 and added to the pre-industrial contribution (Fig. S1 shows the spatial distribution of the anthropogenic AOD perturbation). The spatial distributions of $N_{d,cld}$ and $N_{d,rad}$ are modified using the scaling factor $f_N$ (Eq. 2), which varies spatially with the anthropogenic aerosol. The third and fourth simulations are used to isolate ACI and ARI effects in the present-day atmosphere. In the third simulation, PD$_{ARI}$, extinction from the anthropogenic aerosols are included, but the scaling factor $f_N$ is not applied to $N_{d,cld}$ and $N_{d,rad}$; this isolates ARI effects associated with anthropogenic aerosol. In the final simulation, PD$_{ACI}$, the ACI scaling factor $f_N$ is applied, but aerosol extinction remains at pre-industrial values; this isolates ACI effects associated with anthropogenic aerosol.

Figure 1 shows that the simulated spatial distributions of AOD and $N_{d,cld}$ are consistent with present-day observations. In the PD run, the aerosol perturbations are centred over regions with pronounced industrial emissions of sulfate (South and East

**Table 1.** Description of each simulation.

| Simulation name | ARI characteristics | ACI characteristics |
|---|---|---|
| PI (Pre-industrial) | Aerosol extinction fields for natural aerosol only, represented by the K19 climatology for the year 1850 (K19 only). | Pre-industrial magnitudes of $N_{d,cld}$ and $N_{d,rad}$. $N_{d,cld}$ is spatially constant with a value of 80 cm$^{-3}$. $N_{d,rad}$ follows a vertical profile according to Eq. 1 and varies spatially with $N_{d,rad-sfc}$ set to 120 cm$^{-3}$ over land and 80 cm$^{-3}$ over oceans. |
| PD (Present-day) | Aerosol extinction fields include anthropogenic contribution, represented by the plume model MACv2-SP for the year 2016 (K19 + MACv2-SP). | Present-day magnitudes of $N_{d,cld}$ and $N_{d,rad}$. Global distributions of $N_{d,cld}$ and $N_{d,rad-sfc}$ increased by spatially variable ACI scaling factor $f_N$, as described by Eq. 2. |
| PD$_{ARI}$ (Present-day; isolate ARI) | PD aerosol extinction: global fields of aerosol extinction follow the PD simulation (K19 + MACv2-SP). | PI $N_{d,cld}$ and $N_{d,rad}$. Global distributions of $N_{d,cld}$ and $N_{d,rad-sfc}$ follow the PI simulation. |
| PD$_{ACI}$ (Present-day; isolate ACI) | PI aerosol extinction: global fields of aerosol extinction follow the PI simulation (K19 only). | PD $N_{d,cld}$ and $N_{d,rad}$. Global distributions of $N_{d,cld}$ and $N_{d,rad-sfc}$ follow the PD simulation and are enhanced by $f_N$. |

Asia, North America, and Europe) and biomass burning emissions from agricultural activities in heavily forested regions in the southern hemisphere (South America, South-Central Africa, and the Maritime Continent). $N_{d,cld}$ in the PD run reaches maximum concentrations of about 320 cm$^{-3}$ over East Asia. The spatial distribution and range is consistent with present-day climatologies presented by Grosvenor et al. (2018) and McCoy et al. (2018), who report $N_{d,cld}$ values exceeding 300 cm$^{-3}$

over East Asia and around 200 cm$^{-3}$ off the coasts of the industrial regions of Asia (Fig. 1b). Elevated values are also evident over the Southeast Atlantic Ocean downwind of the African biomass burning regions. A comparison between simulated and observed $N_d$ yields a root mean square error (RMSE) of 49 cm$^{-3}$ and a correlation coefficient of 0.57 (the default parameters $a_N$ and $b_N$ yield an RMSE of 70 cm$^{-3}$ and correlation coefficient of 0.42). The discrepancy is in part due to high simulated values over biomass burning regions that are not reflected in annual mean observations, but also due to regional variability

that MACv2-SP does not capture (e.g., North America). Despite the relatively poor correlation, our idealized representation of aerosols provides appropriate perturbations to the radiative fluxes and bulk cloud properties that are spatially consistent with the dominant sources of global anthropogenic aerosol forcing.

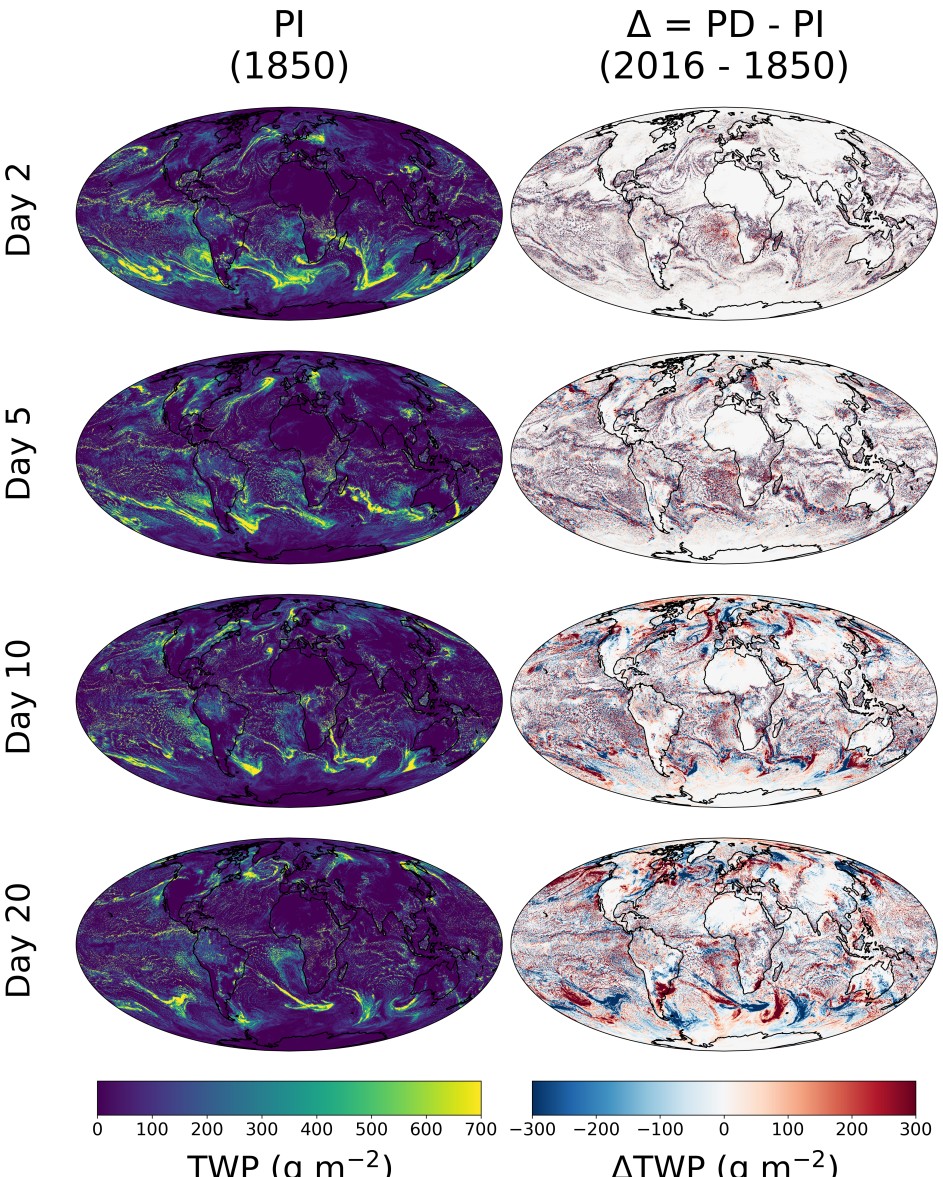

**Figure 2.** Snapshots of day 2, 5, 10, and 20 at 12:00 UTC after the initialization: total water path (TWP) in the PI simulation (left column) and TWP response (PD - PI) to the aerosol perturbation (right column).

Snapshots of the change in total water path (TWP) due to the aerosol perturbation (PD - PI) from four time periods during the simulations are shown in Fig. 2. The limited length of our simulations poses some issues, as it is difficult to disentangle internal variability from the global-scale responses to aerosol effects. By internal variability, we refer to the chaotic nature of the atmosphere, in which small fluctuations grow rapidly in time. For example, Fig. 2 shows that as the simulation progresses

the changes in aerosol concentration have large-scale impacts on the precise timing and location of atmospheric fronts, which appear as a regional change when differencing simulations, but are not usefully considered as a robust 'aerosol effect'. This behavior is similar to initial condition sensitivity where small-scale perturbations at the beginning of the simulation can quickly develop into pronounced changes (Keshtgar et al., 2023; Lorenz, 1963).

Estimating the radiative forcing due to anthropogenic aerosol on a global scale requires multi-year simulations that can robustly separate the response (signal) from internal variability as in e.g. the Coupled Model Intercomparison Project (CMIP) experiments (Schulz et al., 2006). Even longer durations are required to estimate the effective radiative forcing (Boucher et al., 2013). Hence, we do not focus on quantifying the global impact of aerosols on climate and instead focus on the impact of our aerosol perturbation on the regional scale, simultaneously for all regions of the world. We exploit the capability of the model to represent scales that are traditionally used by high-resolution simulations ($\sim 5$ km). These are used by the scientific community to focus on aerosol impacts to radiative fluxes and cloud processes across cloud to regional scales (tens to hundreds of km) and typically run for days to weeks (e.g. Archer-Nicholls et al. (2016); Ban et al. (2021); Che et al. (2021); Dagan et al. (2020); Fan et al. (2013); Marinescu et al. (2021); Heever et al. (2006); Liu et al. (2020); Storer et al. (2010); Takeishi and Wang (2022)). To study aerosol impacts on the global scale we subset the outputs from the global simulations into $15° \times 15°$ regions, producing the equivalent of 288 regional-scale simulations running for a 30 d period. With this method, the regions can interact with each other, and any regional aerosol response is transported to neighboring regions. The power of this configuration is the ability to isolate the different pathways through which aerosols interact with the cloud and atmosphere for a wide range of thermodynamic states and boundary conditions. We can also identify consistent cloud-scale impacts across the globe without the uncertainty (from e.g. different models, parameterizations, schemes, and time periods) that is introduced when traditionally collating simulation data on the spatial variability of aerosol impacts.

## 2.4 Temporal decomposition of regional response

Several recent studies have identified pronounced aerosol effects on clouds and their properties occurring throughout the diurnal cycle (Herbert et al., 2021a; Herbert and Stier, 2023; Hodzic and Duvel, 2018). Therefore, we quantify the regional responses of clouds to the aerosol perturbation over the full diurnal cycle and also the daily mean effect. Data from the 5 km resolution output is re-gridded onto a regular $1°$ grid using the Climate Data Operators (CDO; http://www.idris.fr/media/ada/cdo.pdf, last access: 20 January 2025) software operator *gencon* which generates first-order conservative remapping weights. As we focus on regional domains, we do not lose any information through the re-gridding process. We attempt to isolate the responses due to the aerosol perturbation from internal variability and noise by temporally decomposing the mean time series into short- and long-term components and compositing onto a single diurnal cycle. A seasonal-trend decomposition tool is applied to the response time series (PD - PI) using LOESS (locally estimated scatterplot smoothing) based on Cleveland (1979), providing long-term, short-term, and residual components. LOESS is a statistical decomposition tool that can be applied to extract responses occurring on relatively high frequencies (e.g. diurnal) and has been used in previous climate-focused studies (Deng and Fu, 2019; Carslaw, 2005; Verbesselt et al., 2010; He et al., 2022; Liu and Zhang, 2024; Zhou et al., 2015; Cleveland, 1979; da Silveira Bueno et al., 2024; Papacharalampous et al., 2018; Quan et al., 2016; Jaber et al., 2020; Rabbi and Kovács, 2024;

Moradi, 2022; Deng et al., 2015). Examples of the decomposition for the Congo basin (0°N, 20°E) and the Southeast Atlantic Ocean (10°S, 5°W) are shown in Fig. 3. The short-term component (using an applied periodicity of 1 d) captures aerosol effects on a diurnal timescale, whereas the long-term component captures the internal variability combined with any persistent change. Examples of a persistent change may be a relatively warmer troposphere or enhanced subsidence and may represent an important local or non-local aerosol effect, hence we attempt to recapture this using a second application of the decomposition tool with a prescribed periodicity of 100 d. This provides a time-independent response over the time series, which we attribute to an aerosol effect. This method assumes that any internal variability is evenly distributed around the time-independent response, which may not be true, but provides a reasonable approximation and should capture regions where strong persistent responses occur; we demonstrate our technique using synthetic data in the supporting information (Sect. S2). Recapturing the persistent aerosol effect is well demonstrated in the Southeast Atlantic region (Figs. 3e – h). Here, ACI strongly enhances the LWP of the extensive underlying marine stratocumulus resulting in a persistent positive LWP response with an overlying diurnal cycle. We further reduce the impact of internal variability and noise by compositing the short-term (diurnal) and long-term (persistent) aerosol effects onto a single diurnal cycle.

## 3 Results

### 3.1 Global-scale analysis

In this section, we focus on the regional responses of clouds and radiative fluxes due to the aerosol perturbations across the globe. We focus on regions where we can robustly identify a response, which is achieved by using the following criteria for each variable $X$. To remove regions that have transient synoptic-scale weather (e.g. a mid-latitude cyclone), the regional mean standard deviation of hourly PI values over the time series, $\sum_{\mathrm{hr}=1}^{24} \sigma(X_{\mathrm{PI}})_{\mathrm{hr}} N_{\mathrm{day}}{}^{-1}$, must be within the lowest 50th percentile of the global distribution. This isolates regions that exhibit a consistent diurnal cycle during the PI experiment. The regions where $X$ is likely unimportant are removed when the time series mean in the PI experiment, $\overline{X_{\mathrm{PI}}}$, is in the lowest 25th percentile globally. Finally, we focus our analysis on regions where the response is more pronounced by removing those where the maximum range of the diurnal response, $\overline{\Delta X_{\mathrm{hr\,max}}} - \overline{\Delta X_{\mathrm{hr\,min}}}$, normalized by $\overline{X_{\mathrm{PI}}}$, is in the lowest 10th percentile globally. We determine the dominant driver of the aerosol effect by calculating the RMSE between the responses from the PD and $\mathrm{PD}_{\mathrm{ARI}}$ or $\mathrm{PD}_{\mathrm{ACI}}$ simulations. The difference between the two is used to estimate whether one driver (ARI or ACI) dominates or whether both play a role.

We start by focusing on the responses of LWP (Fig. 4) and precipitation (P; Fig. 5) to the aerosol perturbations. The figures demonstrate considerable spatial variability in the magnitude, direction, and driver of the aerosol effects on clouds.

There is no consistent daily mean regional response in either LWP or P. The percentage increase in the magnitude of $\Delta$LWP varies from 10 – 50 %, with higher values close to or downstream of the aerosol perturbations. The magnitude is not consistently dependent on the aerosol perturbation, which is particularly evident over the Maritime Continent. This is in contrast to $\Delta$P, which tends to be spatially consistent with the aerosol perturbation and of similar magnitudes in all regions (> 45 %). The direction of the change is also inconsistent; $\Delta$LWP tends to be positive over the ocean and negative over the land, whilst $\Delta$P

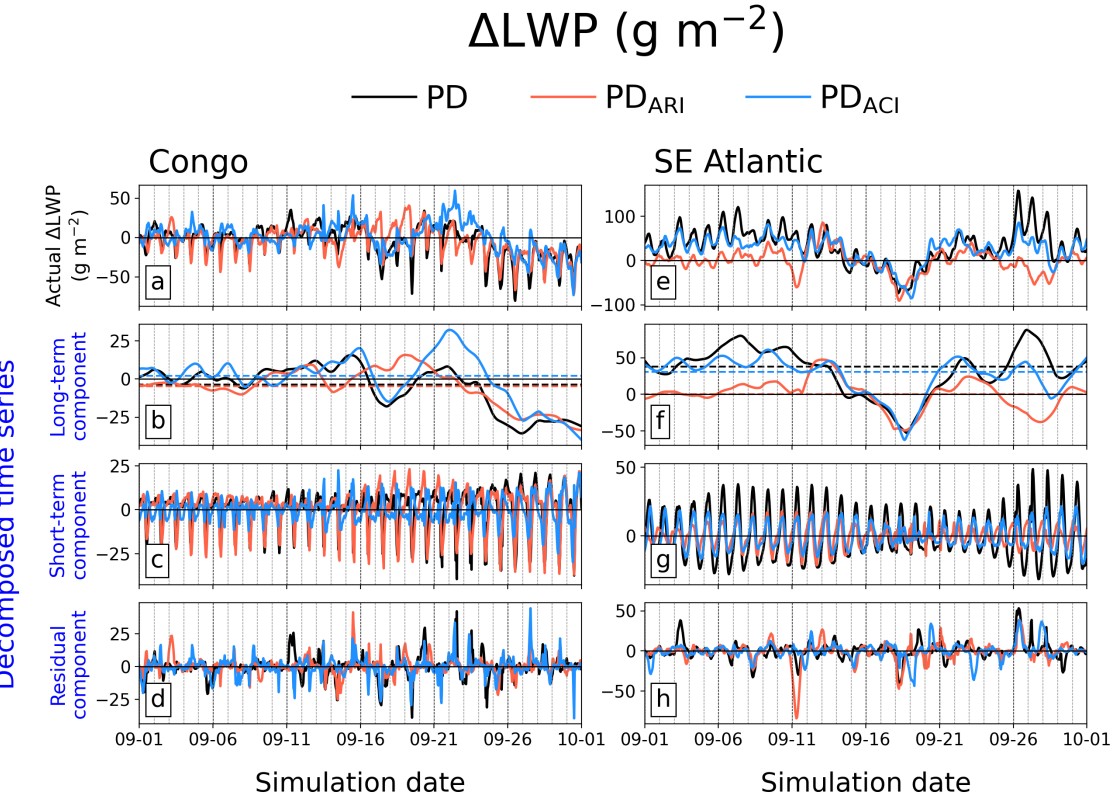

**Figure 3.** Example of the response time series decomposition at two locations: Congo (**a** – **d**) and Southeast Atlantic Ocean (**e** – **h**). The different colors represent ΔLWP (PD - PI) for each PD simulation. The panels show the original time series before decomposition (**a** and **e**), the decomposed long-term component (**b** and **f**), the decomposed short-term component (**c** and **g**), and the residual (**d** and **h**). The horizontal dashed lines (in **b** and **f**) show the persistent aerosol effect that is added to the short-term aerosol effect.

is negative in all regions except the Maritime Continent and West Pacific Ocean. In our model configuration, LWP and P are linked via autoconversion (Sect. 2.1), therefore it is surprising that there is no clear consistency between the responses of the two cloud properties.

Figures 4d – f and 5d – f suggest that the spatial inconsistency in the regional responses is attributable to the lack of consistent underlying aerosol effects. ACI tends to dominate ΔLWP over the ocean, and ARI tends to dominate ΔLWP over land. However, there are only a small number of regions in which the daily mean ΔLWP is fully explained by either of the drivers. Individually, ARI and ACI become more pronounced when ΔLWP is separated into day/night periods, in particular, for ARI in the daytime over Central Africa and East Asia. This suggests that in some regions ACI and ARI are more or less active during different periods of the diurnal cycle. The response of P shows similar behaviour, with both ACI and ARI influencing the daily mean. However, in contrast to LWP, ARI tends to be the main driver of ΔP on the global scale.

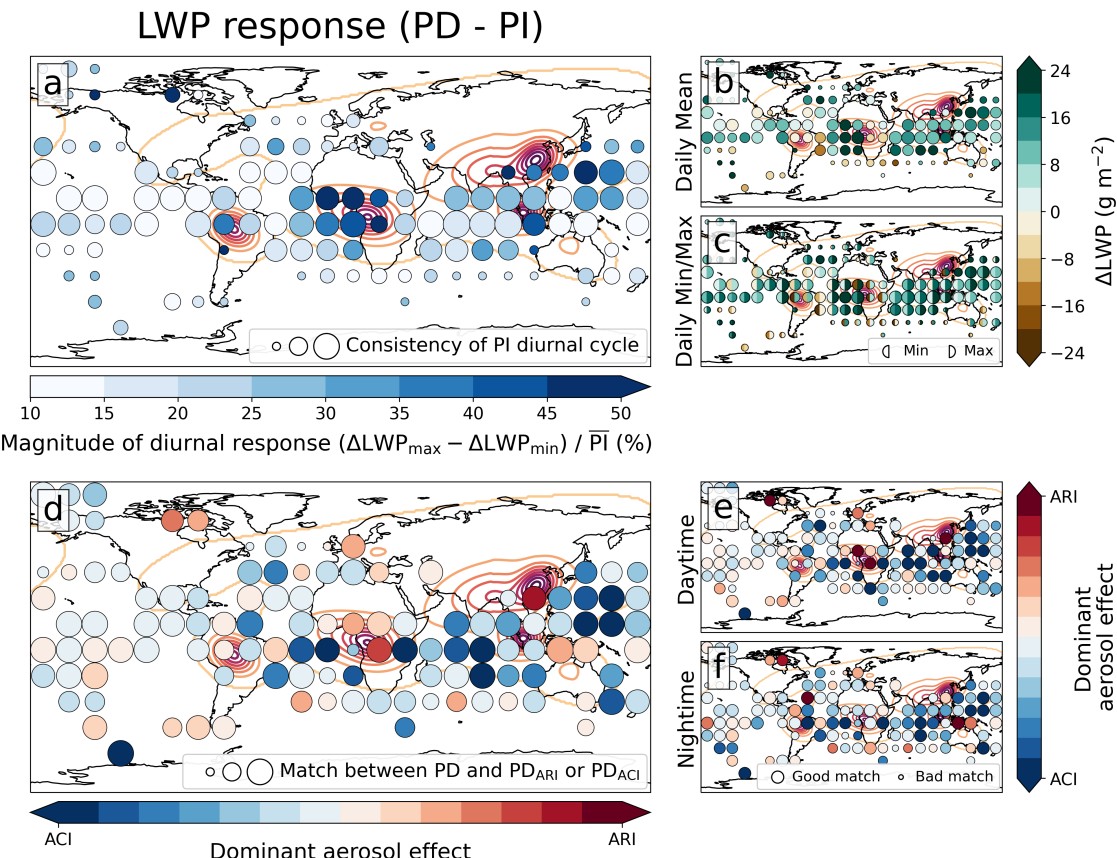

**Figure 4.** Mean diurnal response of liquid water path (LWP) to the aerosol perturbation (PD - PI) from each $15° \times 15°$ region. Panels **a** – **c** show the diurnal magnitude of the response as a percentage (**a**), absolute daily mean (**b**), and absolute daily minimum/maximum (**c**). A larger circle size in **a** – **c** represents a location with an increasingly consistent diurnal cycle throughout the PI simulation. Panels **d** – **f** show the dominating aerosol effect (ARI/ACI) driving the LWP response during the diurnal cycle (**d**), day (**e**), and night (**f**). A larger circle size in **d** – **f** represents a better match between the individual response (PD$_{ARI}$ or PD$_{ACI}$) and total response (PD). All panels show the AOD perturbation as contour lines at 0.05 increments.

One source of spatial consistency is the range of the LWP and P responses during the diurnal cycle. $\Delta$LWP (Fig. 4c) and $\Delta$P (Fig. 5c) range between $\sim$10 – 20 g m$^{-2}$ for LWP and 1 – 2 mm d$^{-1}$ for P. This is a consistent feature in all regions. In some (e.g. the Congo basin and Amazon rainforest) the daily mean response is small but the diurnal range is large, indicating contrasting periods of negative and positive responses during the day. Hence, the daily mean aerosol effect masks the underlying diurnal response. This is further explored in Fig. 6, where min/max $\Delta$LWP and its drivers (ARI/ACI) are shown for the diurnal response.(Figs. 6a – b) and with the addition of the persistent response (Figs. 6c – d). All regions exhibit a marked diurnal range, particularly over land and close to the aerosol perturbations. ARI drives most of the range in the regions closest to the perturbations, which demonstrates that the impact of aerosol on clouds has a diurnal driver that may be dependent on the

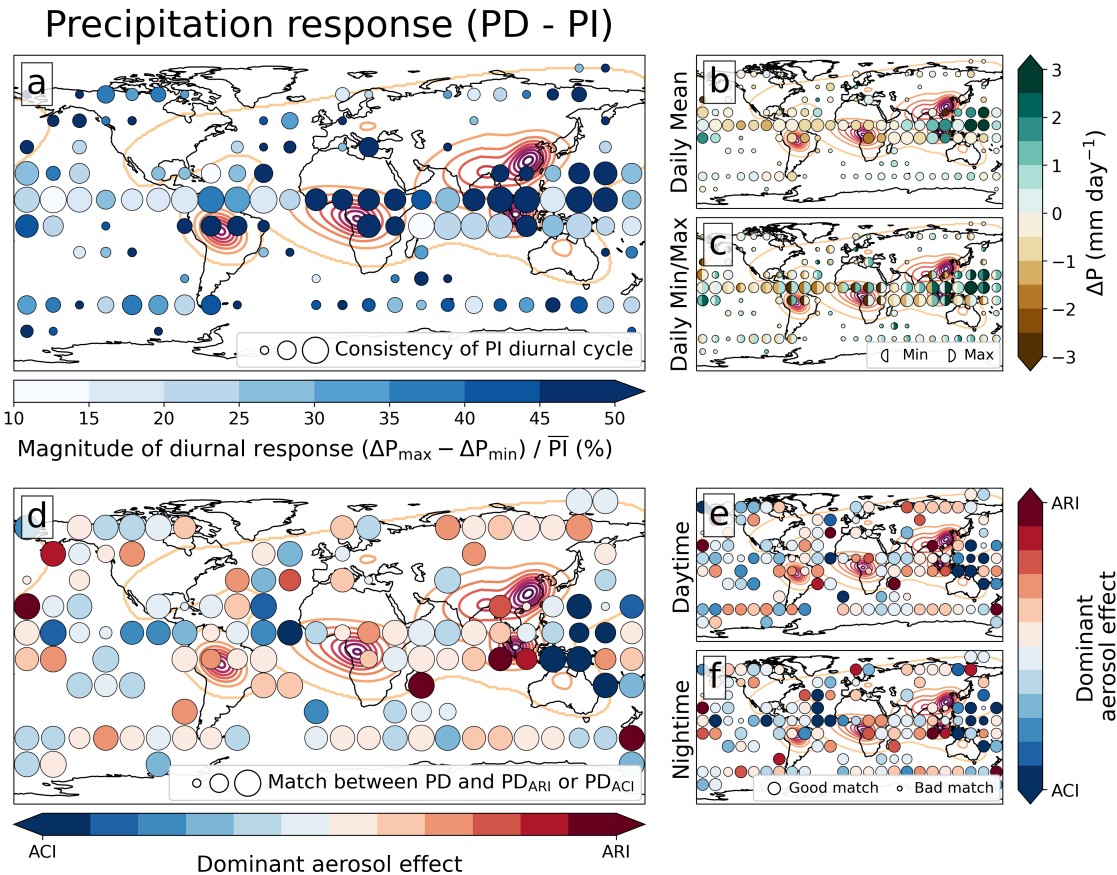

**Figure 5.** Mean diurnal response of precipitation (P) to the aerosol perturbation (PD - PI). Figure details as Fig. 4.

underlying diurnal cycle of clouds, dynamics, or solar radiation. The addition of the persistent response shifts the min/max ΔLWP towards higher magnitudes in all regions due to a strong role from ACI. This suggests that for the aerosol effects on LWP, ARI drives a strong diurnal response, whilst ACI drives an underlying persistent response. The magnitude by which each driver influences ΔLWP explains the spatial variability observed in Fig. 4. We explore the pathways through which ARI and ACI drive the diurnal and persistent responses further in Sect. 3.2.

The diurnal timing of the strongest response of clouds to the PD aerosol perturbation suggests impacts to convective processes over land and enhanced cloud growth in shallow clouds over marine environments. Figure 7 shows the local solar time (LST) at which the maximum absolute response occurs for LWP, ice water path (IWP), P, and cloud condensate mass flux at 500 hPa ($M_{flux}$; calculated on ascending grid points where the vertical velocity at 500 hPa is positive). The maximum ΔLWP occurs during early morning (05:00 – 11:30 LST) over oceans, and in the afternoon (12:00 – 15:00 LST) over land. The former is consistent with peak marine stratus growth (Wood, 2012), while the latter is consistent with the initiation of afternoon convection in the tropics (Worku et al., 2019). The maximum in $\Delta M_{flux}$ (largely limited to regions over land) also occurs during the

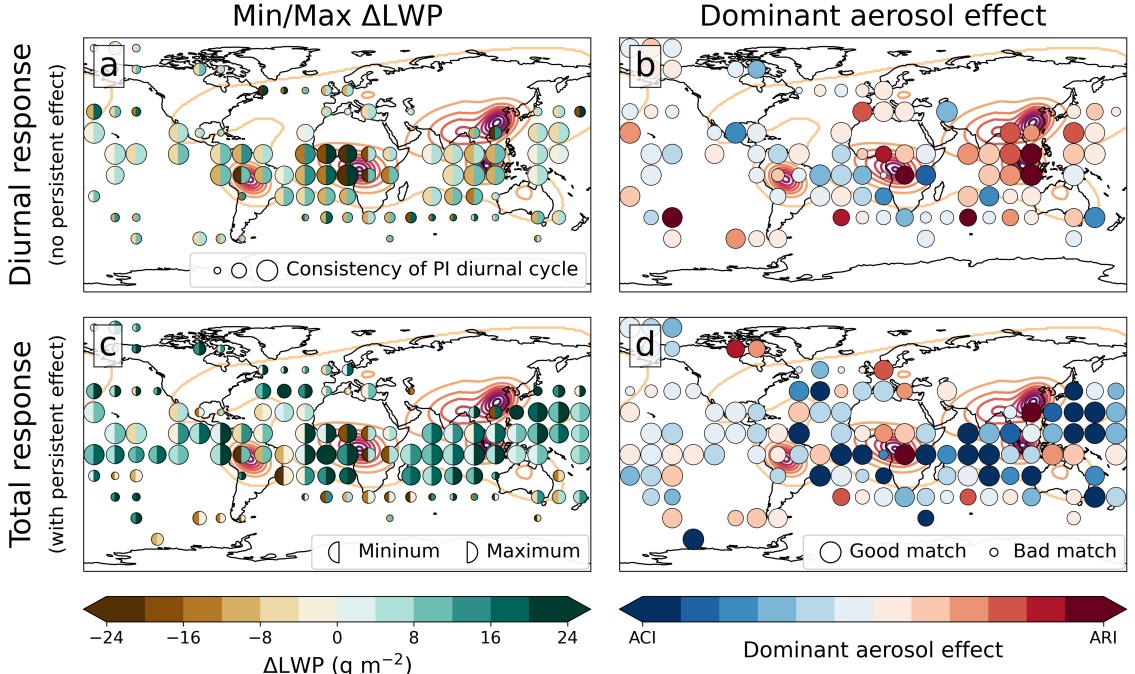

**Figure 6.** Contributions to $\Delta$LWP from the short-term diurnal component (**a** and **b**) and with the addition of the long-term persistent component (**c** and **d**). Panels (**a**) and (**c**) show the minimum (left hemisphere) and maximum (right hemisphere) absolute response during the diurnal cycle with larger circles representing an increasingly consistent LWP diurnal cycle throughout the PI simulation. Panels (**b**) and (**d**) show the dominating process (ARI/ACI) driving $\Delta$LWP, with larger circles representing a better match between the $PD_{ARI}$ or $PD_{ACI}$ and PD.

afternoon, suggesting a link with convection. $\Delta$P and $\Delta$IWP demonstrate similar variability: over land, the maximum occurs
in the afternoon, whereas for a few regions, most noticeably around the Maritime Continent, the maximum occurs overnight
or in the morning. The timing of the maximum responses suggests links to convection over land and to shallow clouds over
marine environments. This is explored further in Sect. 3.2.

The daily mean shortwave (SW) TOA radiative effect due to the aerosol perturbation is similarly region-dependent both in
sign and magnitude (Fig. 8). The cloudy-sky $\Delta SW_{TOA\uparrow}$ drives most of the diversity and is largely correlated with the total
cloud fraction response in Fig. 8a. The magnitude of cloudy-sky $\Delta SW_{TOA\uparrow}$ is sensitive to $\Delta$LWP during the day and the
increase in cloud droplet effective radius (which is positively correlated with aerosol), resulting in enhanced or suppressed
cloudy-sky $\Delta SW_{TOA\uparrow}$ depending on the region. Figure 7 suggests that the cloudy-sky $\Delta SW_{TOA\uparrow}$ will not be directly corre-
lated with the daily mean responses in cloud fraction and LWP due to region-dependent timings of maximum response. The
clear-sky $\Delta SW_{TOA\uparrow}$ is positive in all regions due to the aerosol direct effect (Fig. 8b) and spatially varies with the magnitude
of the perturbation. The all-sky $\Delta SW_{TOA\uparrow}$ is influenced by the clear-sky and cloudy-sky responses and displays considerable
variability. The magnitude and distribution of the all-sky $SW_{TOA\uparrow}$ response is consistent with anthropogenic aerosol radiative
forcing estimates from modelling studies (Fiedler et al., 2019; Myhre et al., 2013; O'Connor et al., 2021).

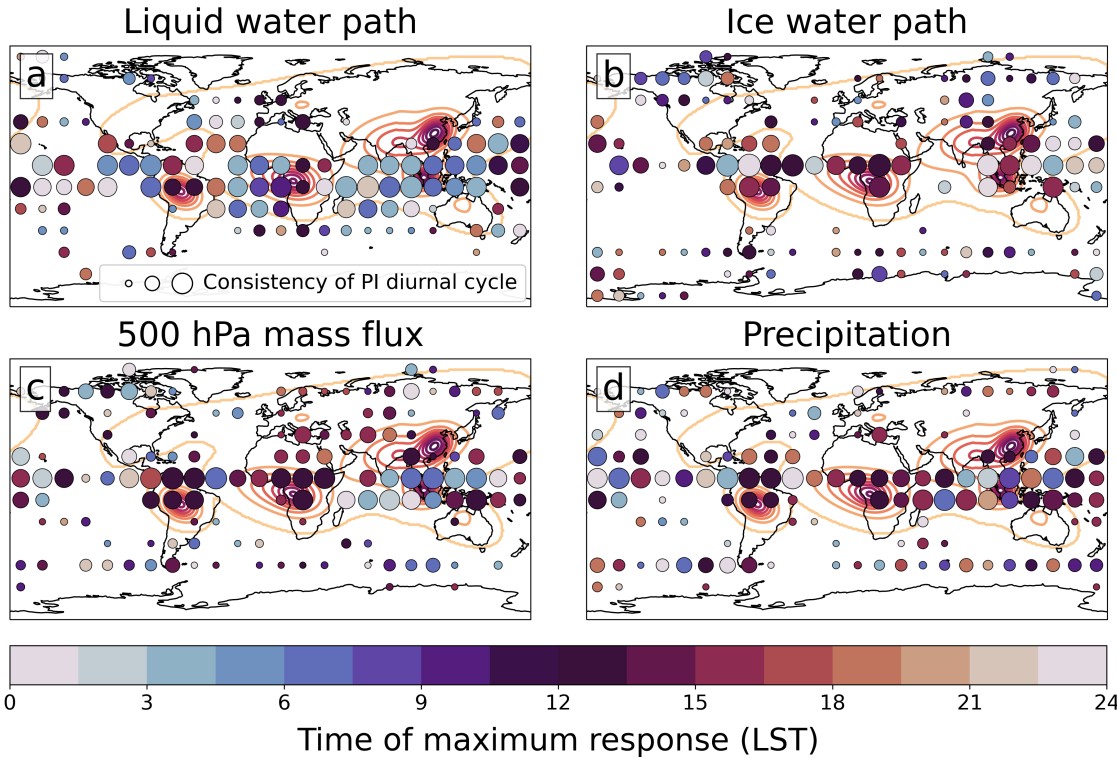

**Figure 7.** Time (LST) during diurnal cycle of maximum absolute response for each $15° \times 15°$ region in the PD experiment. Panels show LWP (**a**), IWP (**b**), M$_{flux}$ at 500 hPa (**c**), and precipitation (**d**). A larger circle size represents a location with an increasingly consistent diurnal cycle of the variable throughout the PI simulation.

The global-scale analysis demonstrates two important results. First, there is considerable spatial variability in the magnitude and sign of the cloud and radiative response to the aerosol perturbation, suggesting aerosol impacts are highly region-specific
and likely dependent on the underlying thermodynamic state of the region, as well as the scale and radiative properties (scattering or absorbing) of the perturbation. Second, despite the variability, there are suggestions of key underlying processes that are regionally independent that may link the aerosol perturbation with the response. We explore this further in the next section.

### 3.2 Regional-scale analysis

In this section, we focus on the regional-scale response of cloud properties and thermodynamic profiles to the aerosol pertur-
340 bation in six regions that demonstrated considerable sensitivity in Sect. 3.1. The spatial domains of the six regions are shown in Fig. 9. We selected three convective regions that play a key role in shaping the tropical large-scale circulation and three regions heavily influenced by our aerosol perturbation. The Amazon rainforest and the Congo basin are characteristic of continental convective regions and are both impacted by localized biomass-burning aerosol. Similarly, the Maritime Continent is impacted by biomass-burning aerosol and deep convection, but situated within the globally important tropical warm pool re-

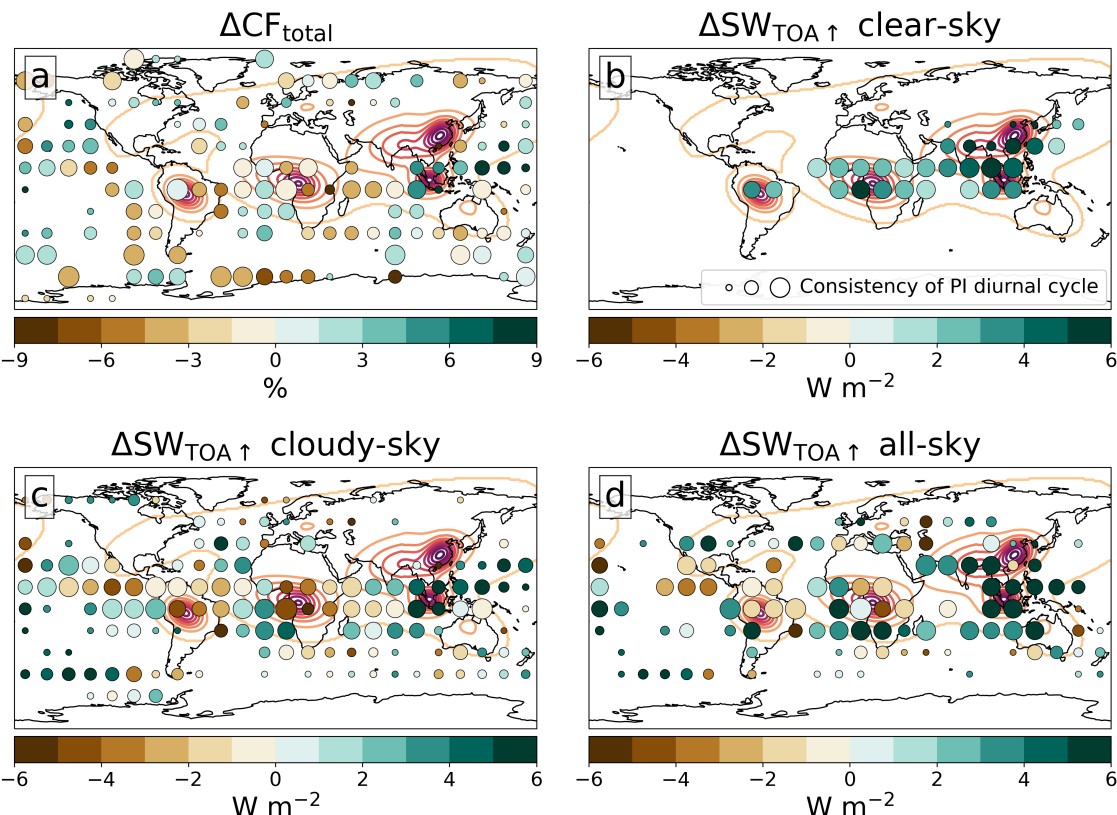

**Figure 8.** Mean diurnal response of clouds and $SW_{TOA\uparrow}$ to the aerosol perturbation from the PD simulation for each $15° \times 15°$ region. Panels show daily-mean $\Delta CF_{total}$ (**a**) and $\Delta SW_{TOA\uparrow}$ in clear-sky (**b**), cloudy-sky (**c**), and all-sky (**d**) conditions. A larger circle size represents a location with an increasingly consistent diurnal cycle of the variable throughout the PI simulation.

gion (De Deckker, 2016). The Southeast Atlantic and Northwest Pacific Oceans are maritime environments situated downwind of regions with strong aerosol perturbations, and East Asia is a continental region with strong localized sulfate emissions. The novel aspect of this study is the globally resolved deep convection (Sect. 2.1), hence we focus primarily on regions associated with deep convection.

### 3.2.1 Response of liquid water path

Figure 10 shows the diurnal change in LWP due to the aerosol perturbation in the six defined regions.

The PD$_{ARI}$ response is consistent with a modification to the large-scale dynamical properties. In the convective regions (Congo, Amazon, Maritime Continent) ARI consistently suppresses LWP between 12:00 and 15:00 LST, temporally consistent with the initiation and evolution of deep convective cells, indicating that ARI from absorbing aerosol suppresses deep convection. This is in agreement with modelling studies over the Amazon (Herbert et al., 2021a; Liu et al., 2020; Martins

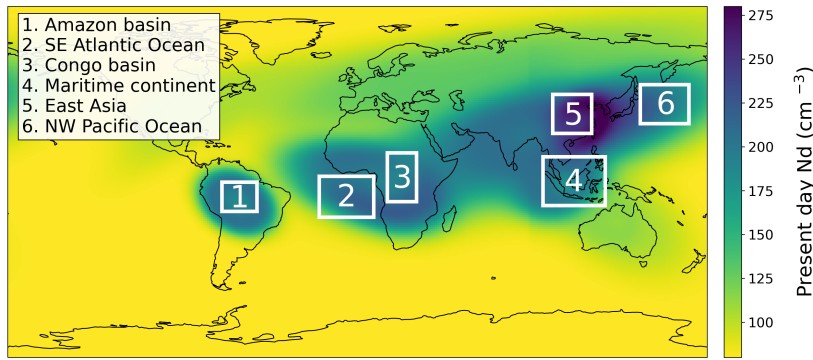

**Figure 9.** Domains used for the regional analysis in Sect. 3.2 outlined by white boxes (labeled 1 – 6). Domains are positioned over regions subject to large aerosol perturbations in the PD simulation as illustrated by $N_{\text{d,cld}}$ shown in the background.

et al., 2009), Indonesia (Hodzic and Duvel, 2018), and Central Africa (Sakaeda et al., 2011). These simulations suggest that in regions dominated by biomass-burning aerosol, ARI consistently impacts deep convection by suppressing activity during the afternoon. However, the magnitude of ΔLWP is region dependent, ranging between -12 % (Maritime Continent), -29 % (Congo), and -33 % (Amazon). This sensitivity correlates with the strength of the aerosol perturbation (Table 2) and the underlying magnitude of the afternoon LWP in the PI experiment. However, this will also be sensitive to the thermodynamic properties of the region that provide the potential for convection (the convective environment) (Williams et al., 2022), the different aerosol plume characteristics, or buffering of the response due to coupling to large-scale meteorology (Stevens et al., 2013). The Maritime Continent includes both land and ocean, so the relatively weaker sensitivity may be associated with the variability in the response over land and ocean (Takeishi and Wang, 2022). The SE Atlantic displays a small LWP suppression during the daytime and enhancement overnight, with an overall negligible daily-mean effect. This is consistent with some studies (Sakaeda et al., 2011; Lu et al., 2018) but not with others that show stronger aerosol sensitivity (Gordon et al., 2018; Che et al., 2021). The marine stratocumulus clouds in this region are known to be sensitive to the vertical structure of temperature, moisture, and biomass burning aerosol (Herbert et al., 2020; Koch and Del Genio, 2010; Wood, 2012), which exhibits more complexity than our idealized aerosol plume. East Asia demonstrates a diurnal cycle in ΔLWP similar to convective regions, which is consistent with modeling studies of the region showing that aerosol suppresses convection due to surface cooling and stabilization of the planetary boundary layer (Liu et al., 2024, 2018). The NW Pacific shows a persistent enhancement of LWP, though this region (and East Asia) are heavily influenced by day-to-day variability of the diurnal cycle (Figs. 10d and e), which limits our ability to isolate the underlying impacts here.

The response of LWP to the aerosol perturbation in the PD$_{\text{ACI}}$ experiment is an overall enhancement observed in all six regions. The continental convective regions show a positive ΔLWP during the day, coinciding with the initiation of deep convection. This is consistent with Herbert et al. (2021a) who showed that deeper clouds, with greater condensate loading, are more sensitive to ACI. Over the Maritime Continent, there is a persistent enhancement in ΔLWP throughout the diurnal cycle

# Liquid water path

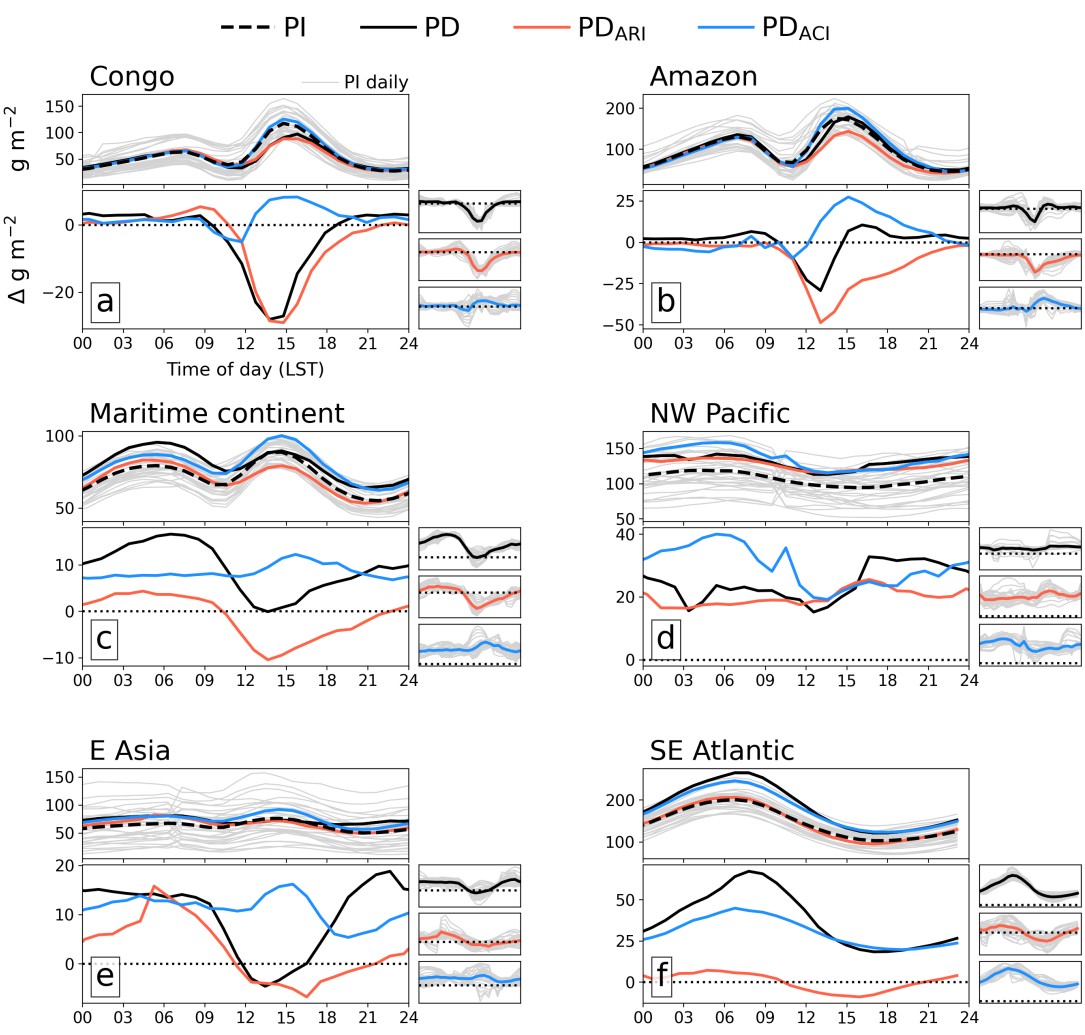

**Figure 10.** Composites of the decomposed LWP diurnal cycle and its response to the aerosol perturbation over the six regions of interest. For each region (**a** – **f**) the top sub-panel shows the mean diurnal cycle of LWP ($\mathrm{g\,m^{-2}}$) in each simulation, with grey lines showing each day of the PI simulation. The lower sub-panels show $\Delta$LWP from each PD simulation (PD$_X$ - PI), which are repeated individually to the right along with grey lines showing each day of the $\Delta$LWP composite.

of +10 %. This reflects the prevalence of low-level marine clouds over much of this region and is associated with enhanced cloud cover (Figs. 8a and S5) and evaporation from the ocean surface (Fig. S6). The Congo and the Amazon do not have this persistent enhancement, reflecting the dominance of deep convection in driving the diurnal cycle of LWP. The SE Atlantic, 

characterized by widespread low-level stratocumulus, displays a strong and robust persistent enhancement of LWP due to ACI

**Table 2.** Mean AOD of each domain and change in AOD due to the PD aerosol perturbation alongside the mean value of $f_N$.

| Region (label of Fig. 9) | $AOD_{1850}$ | $\Delta AOD_{2016-1850}$ | $f_N$ |
|---|---|---|---|
| Amazon basin (1) | 0.14 | 0.38 | 3.7 |
| SE Atlantic Ocean (2) | 0.18 | 0.19 | 2.1 |
| Congo basin (3) | 0.26 | 0.35 | 2.3 |
| Maritime Continent (4) | 0.09 | 0.25 | 3.7 |
| East Asia (5) | 0.13 | 0.44 | 4.4 |
| NW Pacific Ocean (6) | 0.08 | 0.06 | 1.7 |

reaching +25 % at night with very little day-to-day variability. The positive relationship between $\Delta$LWP and $\Delta N_{d,cld}$ here is consistent with remote sensing observations from Michibata et al. (2016), but inconsistent with those from Sato et al. (2018), and may be sensitive to the representation of the warm-rain process (Gryspeerdt et al., 2019; Sato et al., 2018; Terai et al., 2020); we revisit this in the conclusions. East Asia and the NW Pacific show similar persistent enhancements of LWP though there is considerable day-to-day variability.

When the ARI and ACI effects are isolated in the $PD_{ARI}$ and $PD_{ACI}$ simulations, the cloud LWP responses are consistent across the six regions. However the combined effect in the PD simulation is not, suggesting region-dependent nonlinearity between ARI and ACI. This is consistent with the results in Sect. 3.1. Over the Congo, $\Delta$LWP is driven by ARI with very little role from ACI, whereas over the Amazon $\Delta$LWP is largely a linear combination of the isolated aerosol effects. In the Amazon region, the timing of the isolated ACI response suggests that ARI is driving reductions in deep convection, but ACI is impacting the resulting properties of the clouds that form - thereby explaining the overall $\Delta$LWP. Herbert et al. (2021a) reported regime-dependent responses of convective clouds to aerosol, with ACI evident in shallow cumulus and ARI evident in deeper clouds. Liu et al. (2024) also found contrasting aerosol impacts to the shallow and deep convective regimes over East Asia, and Sheffield et al. (2015) found that ACI was primarily active in cumulus congestus clouds. In the Maritime Continent, $\Delta$LWP is a linear sum of ARI and ACI during the afternoon, but is nonlinear overnight into the morning. The SE Atlantic region also displays nonlinearity, with $\Delta$LWP in the PD simulation greater than the sum of the two aerosol effects, most evident during the morning. The similarities of the two regions point to nonlinearity occurring in the shallow marine clouds and may be associated with a positive feedback. The NW Pacific and East Asia regions show nonlinearity in the opposite direction, with the $\Delta$LWP less than the sum of the two aerosol effects. However, given the natural variability here, it is not possible to say whether this is an appropriate conclusion.

### 3.2.2 Response of convection and cloud vertical profiles

In Figures 11 – 13 we focus on the drivers of the cloud response to aerosol perturbations in the three convective regions. Variables include IWP, P, and $M_{flux}$, and profiles of ice water content (IWC), liquid water content (LWC), potential temperature ($\theta$), water vapor (Qv) and vertical velocity (W*) calculated in regions characterized by ascent (1° grid boxes where the mean

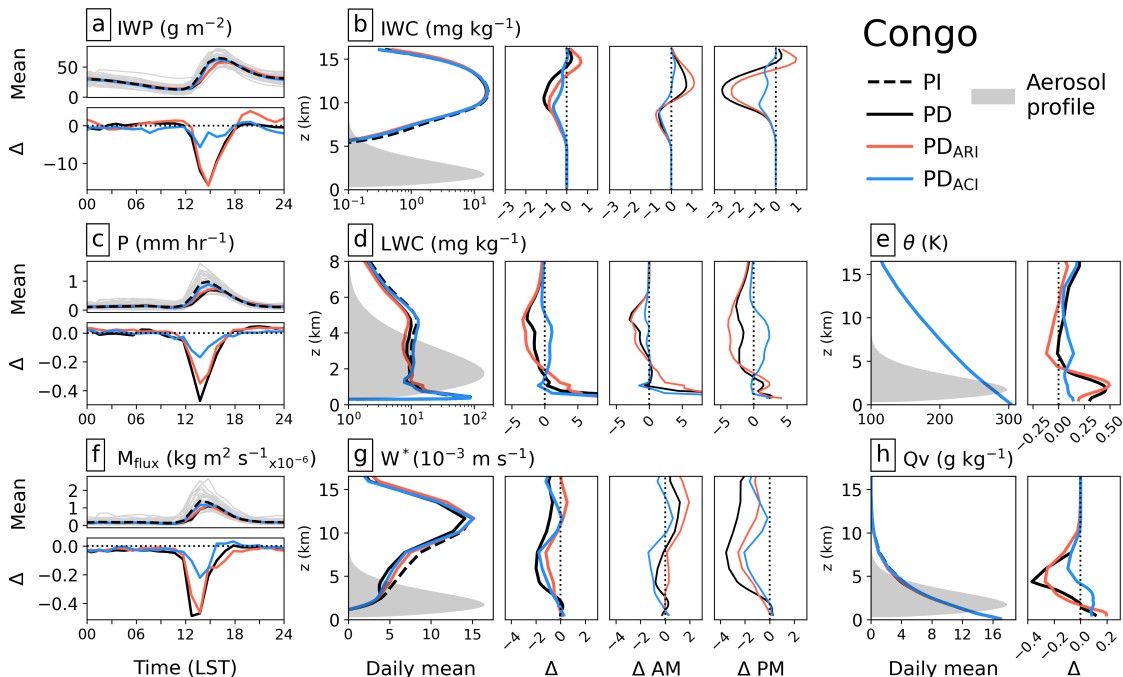

**Figure 11.** Composites showing the regional-mean change in cloud and thermodynamic properties in the Congo region. Diurnal composites (**a**, **c**, **f**) show mean diurnal cycles of IWP (**a**), precipitation rate (**c**), and M$_{flux}$ (**f**) in the top sub-panel, and the response of each variable to the aerosol perturbation in the lower sub-panel (PD$_X$ - PI). Mean vertical profiles are shown for IWC (**b**), LWC (**d**), potential temperature $\theta$ (**e**), vertical velocity W* (**g**), and water vapor Qv (**h**). Profiles for each variable include the mean from each simulation on the left and diurnal-mean changes due to the aerosol perturbation on the right (PD$_X$ - PI). Plots **b**, **d**, and **g** also show the diurnal-mean change separated into contributions from the AM (00:00 to 12:00 LST) and PM (12:00 - 24:00 LST). Profiles of the aerosol perturbation are shown in grey alongside the mean profiles. Note that the LWC is shown from 0 to 8 km and all other profiles are shown from 0 to 16 km.

vertical velocity at 300 hPa during the PI simulation is positive). The frequency of output on all vertical levels is insufficient (3 hr) to robustly decompose the time series following Sect. 2.4, hence the profiles will include influence from internal variability. To minimize this, the regional-mean responses are composited onto a single diurnal cycle. Additionally, the limited day-to-day variability evident in Fig. 10 for the regions provides confidence that the responses are primarily due to the aerosol perturbation.

The Congo (Fig. 11) and Amazon (Fig. 12) regions display strong similarities in ΔLWP due to the aerosol perturbation (Figs. 10a – b). In both regions ARI suppresses afternoon convection, reducing the production of condensate and the vertical extent of the deep convective clouds. M$_{flux}$ is reduced by 30 % (Congo) and 20 % (Amazon) with weakened W* throughout the column in the afternoon (PM) period. The strongly absorbing aerosol produces localized heating of the smoke layer, suppressing mixing in the lower atmosphere and drying aloft, which reduces the potential for convection in the region. The suppressed convection reduces the regional-mean vertical extent of clouds and decreases LWC throughout the column. This is consistent with other studies over Central Africa (Sakaeda et al., 2011) and South America (Liu et al., 2020; Koren et al., 2008; Thornhill et al.,

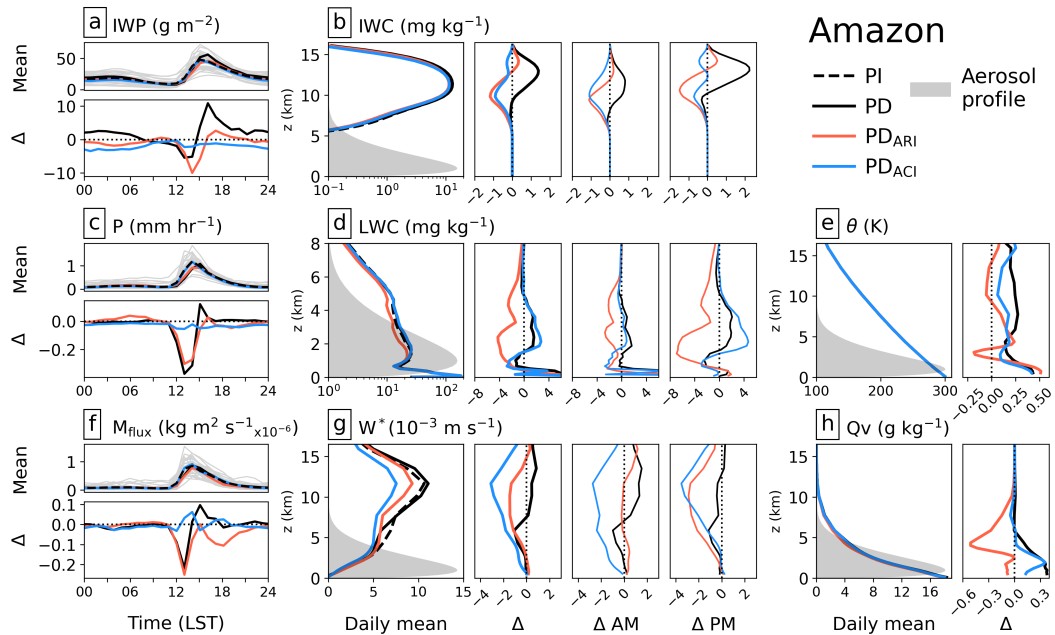

**Figure 12.** Same as Fig. 11 but for the Amazon region.

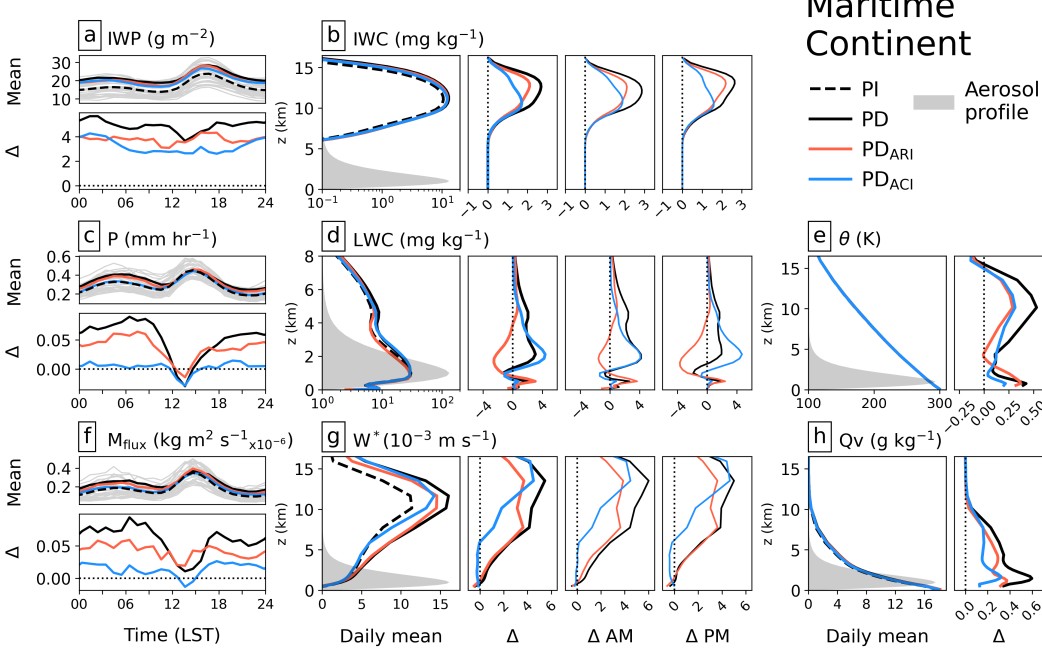

**Figure 13.** Same as Fig. 11 but for the Maritime Continent region.

2018). A similar change in AOD over the two regions (Table 2) results in a comparable suppression in $M_{flux}$ ($\sim$ 30 and 20 %). This is consistent with the findings of Herbert et al. (2021a). However, the percentage change of the $\Delta W^*$ profile and $\Delta$LWP is greater in the Amazon and suggests the differences may be due to a stronger capacity to buffer the perturbation over the Congo, which tends to exhibit more convection than the Amazon, or differences in the convective environments (Storer et al., 2010) that can result in one region being more susceptible to the aerosol perturbation. Changes to convection and the vertical transport of condensate strongly suppresses IWP during the afternoon, with a smaller enhancement during the evening. This is consistent with Herbert et al. (2021a) who found that absorbing aerosols over the Amazon caused the accumulation of convective available potential energy (CAPE) to be released later in the afternoon, driving some convection yet not to the full extent as without the presence of aerosols. The ACI pathway drives a redistribution of liquid water in both regions towards the top of the deep clouds. Positive $\Delta$LWP and negative $\Delta P$ for $PD_{ACI}$ in the afternoon is consistent with suppression of the warm rain process. In both regions ACI increases LWC in the lowest 1 km and suppresses vertical ascent. Some modelling studies have suggested aerosols can also directly influence convection through invigoration of convective cloud cores via ACI, either in the liquid phase (Lebo, 2018; Sheffield et al., 2015; Fan et al., 2018) or ice phase (Heever et al., 2006; Fan et al., 2013), whilst others report suppression or regime-dependence Khain et al. (2008); Lebo and Seinfeld (2011); Storer et al. (2010); Igel and van den Heever (2021). This uncertainty is consistent with the model intercomparison of Marinescu et al. (2021).

The Congo and Amazon regions respond consistently to the aerosol perturbation when ARI and ACI effects are considered in isolation, but the combined effect differs, suggesting a degree of thermodynamic state dependence. In the Congo region (Fig. 11) the responses of many variables have largely additive contributions from ARI and ACI (e.g. $\theta$, $W^*$, IWC, LWC above 2 km, and P), with ARI tending to drive stronger regional responses than ACI. In contrast, the Amazon region (Fig. 12) does not consistently show additive aerosol effects. Some variables show largely additive responses (e.g. LWP, precipitation, $W^*$ below 7 km) driven primarily by ARI but others are not clearly attributable to either ARI or ACI (e.g. IWC, $W^*$ and $\theta$ above 7 km). In contrast to the Congo region, ACI plays a stronger role in the Amazon and is responsible for most of $\Delta$LWP and $\Delta$LWC, but only weakly impacts P. The enhanced role of ACI in the Amazon is consistent with relatively higher frequency of shallow convection than in the Congo, which is a regime known to be sensitive to ACI (Langton et al., 2021; Sheffield et al., 2015). The contrasting roles of ACI and ARI in the Congo and the Amazon suggest that the response of convection to changes in the aerosol population is dependent on the background thermodynamic state and convective environment, which has also been observed in remote-sensing studies of the Amazon region (Herbert and Stier, 2023; Ten Hoeve et al., 2011; Yu et al., 2007). This is also consistent with Chang et al. (2015) who show that the sensitivity of deep convective clouds to aerosols is regime-dependent due to nonlinearity between dynamical and microphysical processes.

The Congo and Amazon are strongly perturbed by aerosol from biomass burning sources. The primary driver is a localized modification to the convective environment that suppresses convection and reduces daily accumulated P by 1 mm day$^{-1}$ in both regions ($\sim$ 15 % and 10 % of PI values for Congo and Amazon). The P response is associated with ascending regions (Fig. S7 and S8), linking the changes to convection. This is consistent with Barkhordarian et al. (2019) who report a long-term drying of the Amazon partially driven by changes to cloudiness and precipitation patterns they associate with biomass burning aerosol. Long-term trends are not observed over the Congo region, but decadal-scale P trends in Western Africa have been

shown to be sensitive to aerosol in GCMs (Zhang et al., 2021). This study suggests the modification to deep convection may have an additional impact on P over the region which is unlikely to be represented in GCMs. Additionally, non-local sources of moisture have been found to be important in driving convective activity in these regions (Creese and Washington, 2018; Wu and Lee, 2019), which suggests that both scales (convection permitting resolution and large-scale drivers) need to be represented to fully capture the impact of aerosol perturbations and greenhouse gases on P trends over continental convective regions.

The response of the cloud field over the Maritime Continent (Fig. 13) is consistent with the other convective regions, but in addition, the aerosol perturbation impacts the large-scale circulation. A persistent aerosol effect in this region (and general absence over land) was identified in Fig. 6. This is evident in the decomposed diurnal cycles of $\Delta$LWP (Fig. 10c) and $\Delta$P (Fig. 13c) that exhibit a largely time-independent response combined with an additional response in the early afternoon. The latter is consistent with the impacts to the convective environment as observed over the Amazon and the Congo, while the former is a modification to the large-scale circulation. ARI drives a persistent positive $\Delta M_{flux}$ of $\sim 20\%$, primarily due to enhanced ascent throughout the column (Fig. 13g). The response of the large-scale circulation due to ARI is consistent with the strengthening of the Walker Circulation and tropical ascent reported by Williams et al. (2022), where the anomalous source of diabatic heating projects onto the ascending branch of the Walker Circulation. The global-scale analysis of $\Delta M_{flux}$ in Fig. S3 shows a negative response over the Western Indian Ocean, which supports this hypothesis. ACI also drives a persistent increase in W*, but only above 6 km. This occurs alongside an increase in IWP and $\theta$, which suggests a role for direct modification of the convective cloud cores via convective invigoration from the cold phase (Heever et al., 2006; Fan et al., 2013). This was not observed over the Congo or Amazon, which is consistent with Khain et al. (2008) who found that convective invigoration occurred in moist maritime deep convection but did not in drier, continental, deep convection. The sensitivity of deep convection in this region is consistent with regional modelling studies (Lee and Wang, 2020; Takeishi and Wang, 2022; Chang et al., 2024), yet there is no agreement on the sign or magnitude of the response. Chang et al. (2024) demonstrate that it is likely linked to the large-scale convective environment influenced by El Nino conditions. However, none of these studies report a persistent increase due to changes in the large-scale circulation, which may be due to the inability of these model configurations to represent the large-scale dynamical feedback that we simulate.

The overall response of the Maritime Continent to the aerosol perturbations is driven by both ARI and ACI, with some properties of the cloud and atmosphere dominated by one of the pathways. The diurnal cycles of LWP, $M_{flux}$, and P are approximately a linear sum of the contributions from the two pathways. During the AM period the LWC profile response is controlled by ACI effects, whilst during the PM time period both ARI and ACI contribute to the changes - illustrating the connection between ARI and deep convection over the land. ARI dominates the response of W* in the warm-phase regions of the cloud (up to 5 km), whilst both ARI and ACI are active in the ice-phase regions. This highlights that the Maritime Continent may be particularly sensitive to anthropogenic aerosol due to its position within the Walker circulation and the pronounced diurnal cycle of convection. It is possible that other regions within ascending or descending branches of global atmospheric circulation may exhibit similar sensitivity (Williams et al., 2022) and should be considered in future studies.

## 4   Conclusions

In this study we make the first steps towards investigating the impact of anthropogenic aerosol on clouds, precipitation and radiation in a global kilometer-scale configuration of the ICON model. We focus on the rapid response of cloud and climate to a prescribed global aerosol perturbation, which we represent using the MACv2-SP plume model. We ran simulations for the month of September, both for pre-industrial (1850) and present-day (2016) aerosol distributions, providing a realistic range of anthropogenic perturbations across the globe. Additional PD simulations were run to isolate the role of ARI and ACI.

In an effort to isolate the aerosol impacts from internal variability, we subset the globe into defined regions and temporally decompose the time series into diurnal and persistent components, which we composite onto single diurnal cycle. In a global-scale analysis we subset the global simulation outputs into 15° x 15° regions, producing the equivalent of 288 regional high-resolution simulations that can interact with each other. We then focus on the regional-scale response at a process level in six locations heavily influenced by the aerosol perturbation.

The global-scale analysis demonstrates considerable spatial variability in the magnitude, direction, and driver of the cloud responses to aerosol perturbations. A focus on $\Delta$LWP and $\Delta$P shows no consistent daily-mean regional response, and whilst $\Delta$P correlates with the aerosol perturbation, $\Delta$LWP does not consistently. The spatial variability is consistent with ARI and ACI effects playing region-dependent roles that are sensitive to the regional thermodynamic environment. Regional responses are rarely fully explained by one pathway, suggesting ARI and ACI both contribute to the total aerosol effect and must both

be taken into account. The spatial variability in how clouds respond to the aerosol perturbation results in associated variability in the TOA shortwave radiative forcing. We have simulated the month of September when biomass burning emissions peak; therefore, we anticipate the spatial distribution of the forcing to be sensitive to the annual cycle.

The sensitivity of $\Delta$LWP to aerosol consistently includes a diurnal component, which may be masked by the daily-mean response. The diurnal range in $\Delta$LWP was greatest over land and close to the aerosol perturbation, demonstrating that the

impact of aerosols on clouds has a diurnal driver that may be dependent on inherent regional diurnal cycles of clouds, dynamics, or solar radiation. The LWP response also includes a persistent increase that was stronger over oceans than on land. On the global scale, and for the regions that we could isolate a response from the aerosol perturbation, ARI tended to dominate $\Delta$LWP on the diurnal cycle and ACI dominated the persistent LWP increase. The pronounced diurnal cycle in LWP sensitivity to aerosol suggests that polar-orbiting remote-sensing platforms, such as those on the A-Train constellation, may struggle to

estimate climate-relevant responses of clouds and climate to aerosol as they only observe a limited period of the diurnal cycle at any one latitude.

A focus on regions impacted by the aerosol perturbations shows some consistent process-level responses. Three regions, characterized by deep convection and emissions of biomass burning aerosol, consistently demonstrated a suppression of the diurnal cycle of convection via modifications to the convective environment due to ARI and enhanced LWP due to ACI.

However, the combined effect (ARI + ACI) differed in each region. The direct modification to convective clouds (suppression or invigoration) via ACI also differed between regions. We hypothesize that the differences are a result of the large-scale thermodynamic environment unique to each region, manifesting as thermodynamic state dependence in the response to the

aerosol. Large-scale responses were evident in the Indo-Pacific warm pool region in the ascending branch of the Walker circulation, driving persistent changes to the large-scale circulation alongside the diurnal cloud-scale response. The global-scale and regional-scale analyses point towards strong regional dependence in the impact of aerosols on clouds and climate, hence the outcomes from isolated case studies are likely not representative for other regions. The results also strongly suggest that ACI and ARI cannot be considered independently as the cloud responses via each pathway do not tend to be additive. Some were dominated by either ACI or ARI, and some behaved nonlinearly, resulting in a combined aerosol effect at odds with the individual components.

In regions not directly influenced by the aerosol perturbation (e.g. remote regions like the Arctic) the decomposition method is unable to sufficiently isolate the cloud responses from internal variability. An extension of this analysis to the entire globe could be achieved via longer simulations (e.g. Bolot et al. (2023); Cheng et al. (2022); Sato et al. (2018)) or ensembles (e.g. Deser et al. (2020); Dittus et al. (2020)). However, this will require considerable computing resources. Sato et al. (2018) ran a year-long global kilometer-scale simulation using the NICAM model and analysed ACI by focusing on the global relationship between LWP and the aerosol number concentration, removing the need to run multiple simulations. An alternative is to nudge the simulation to observed meteorology (e.g. Atlas et al. (2022, 2024); Terai et al. (2020)). However, this will suppress any large-scale modifications, which our results suggest may be an important feature in some regions.

The idealized representation of aerosol and $N_d$ in this model has helped identify important process-level interactions and provides a platform for future studies using realistic aerosol perturbations. The use of non-interactive aerosol may mask important feedbacks and processes including the impact of clouds and precipitation on the spatio-temporal distribution of aerosols, changes to the surface properties and energy fluxes, and turbulence that would influence emissions and aerosol removal processes. Changes in aerosol concentrations would also affect $N_d$ concentrations and vertical profiles. Aerosol emissions also exhibit diurnal cycles (Yu et al., 2021; Torres and Ahn, 2024) that we do not account for.

Future studies should also consider building on the temporal decomposition method (Sect. 2.4) as not all internal variability can be isolated from the aerosol-driven response. The method assumes that mean internal variability during the time series is equal to zero; whilst this may be true on very long time scales (years to decades) it is unlikely to be the case over our simulation duration. The method additionally assumes that the persistent response due to the aerosol perturbation is independent of time. In reality, this component may increase or decrease during the simulation due to local or non-local feedbacks between clouds, the surface, and the thermodynamic properties of the region. This could be explored in future studies with longer simulations.

Additional sources of uncertainty arise from the cloud microphysics scheme and unresolved convection. The choice of cloud microphysics scheme and representation of cold-phase processes have been shown to impact the sensitivity of convective clouds to aerosol (Heikenfeld et al., 2019; White et al., 2017; Sullivan and Voigt, 2021; Marinescu et al., 2021), while the representation of the warm-rain process and its link to aerosols have been shown to be important for ACI impacts on warm-phase clouds (Gryspeerdt et al., 2019; Sato et al., 2018; Terai et al., 2020). Archer-Nicholls et al. (2016) and Possner et al. (2016) have shown that the magnitude of ACI and ARI impacts on clouds may be sensitive to unresolved convection at 5 km resolution, potentially requiring a finer global resolution (e.g. Wedi et al. (2020)). A key reason for the model and microphysics uncertainty is the lack of observational constraints for cloud microphysical processes, particularly in convective updrafts (John-

son et al., 2015; Pathak et al., 2020; Proske et al., 2023). These will be required for evaluating and developing future global kilometer-scale simulations of aerosol-climate interactions. Intensive field campaigns targeting aerosol-convection interactions such as the Tracking Aerosol Convection interactions ExpeRiment (TRACER) and associated campaigns (e.g. Lappin et al. (2023)), will provide valuable observations, and will complement previous field campaigns (e.g. GoAmazon (Martin et al., 2016), ACRIDICON-CHUVA (Wendisch et al., 2016), and CACTI (Varble et al., 2021)). However, there is a lack of intensive field-campaign observations from the convective regions of Africa and Southeast Asia. Existing remote-observation platforms will soon be joined by ESA's Earth Cloud Aerosol and Radiation Explorer (EarthCARE; Illingworth et al. (2015)) and NASA's Plankton Aerosol Clouds and Ecosystems (PACE; Gorman et al. (2019)) satellite. These new missions, focusing on aerosols and clouds, will be a useful addition and help continue the long-term observational record of aerosols in the earth system.

*Code and data availability.* The ICON model is available under an open source (BSD-3C) licence (https://www.icon-model.org) and publicly available at https://gitlab.dkrz.de/icon/icon-model. Our specific code commit is SHA 558a9611f4de09cbd8d46ff4cd2e927b243cf58f. The MACv2-SP software is implemented in the ICON model source code and is publicly available in the supplementary material of Stevens et al. (2017). Full access to the simulation output data and necessary processing resources is available upon request via the DYAMOND initiative website (https://www.gewex.org/dyamond/). All scripts used to analyze the simulation output and produce the manuscript figures are available on the Zenodo repository (Herbert, 2024, https://zenodo.org/records/11470778).

*Author contributions.* PS conceptualized the research. RJH, AILW, PW, DWP and PS designed the methodology and experiments. RJH, AILW, and PW prepared and ran the simulations. RJH and PW processed and analysed the simulation output. RJH prepared the manuscript with contributions from all co-authors.

*Competing interests.* At least one of the (co-)authors is a member of the editorial board of Atmospheric Chemistry and Physics.

*Acknowledgements.* This research was supported by the European Research Council project RECAP under the European Union's Horizon 2020 research and innovation programme (grant no. 724602) and by the FORCeS project under the European Union's Horizon 2020 research programme with grant agreement no. 821205. PS additionally acknowledges funding from the European Union's Horizon 2020 project nextGEMS under grant agreement number 101003470 and the European Union's Horizon Europe project CleanCloud with grant agreement 101137639 and its UKRI underwrite. AILW acknowledges funding from the CIMES Postdoctoral Fellowship under award NA18OAR4320123 from the National Oceanic and Atmospheric Administration, U.S. Department of Commerce. We thank the German Climate Computing Center (DKRZ) for use of its computer facilities on which the simulations were performed.

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
