# Peer review of "Regional variability of aerosol impacts on clouds and radiation in global kilometer-scale simulations"

_EGUsphere, 2024_

## Author Response (AR1)

**Response to reviewers by Herbert and co-authors.**

We thank the three reviewers for providing comments and suggestions. There were several key comments that were consistent amongst the reviewers. We are confident that we have addressed all of these in the revised manuscript. We outline the major revision that has been made to the manuscript first then provide a point-by-point response to each comment.

Following suggestions by all reviewers we have made a major revision to the manuscript in order to better focus the manuscript and make better use of the novel elements of the experiments. The section on the global responses due to the aerosol perturbation has been removed and replaced with new analysis that uses the temporal decomposition method used for the regional analysis. We divide the globe into 15° x 15° regions and temporally decompose the response of cloud properties, precipitation, and radiation over the 30-day time series to obtain a composited diurnal response. With this method we can mitigate or reduce the role of internal variability and provide a new global-scale perspective on how anthropogenic aerosols impact clouds in our convection-permitting model (an example of the analysis is shown below in Fig. R1). In the new section we show the mean diurnal responses of LWP, precipitation, and TOA shortwave radiation on the global-scale. We also show the spatial distribution of the dominating process (ARI or ACI). This new section nicely frames the focus for the remaining results section and helps to strengthen the conclusions.

[Figure]

**Figure R1.** Mean diurnal response of LWP to the aerosol perturbation in the PD experiments from each 15° × 15° region. Panels **a** – **c** show the magnitude of the response as a percentage (**a**), absolute daily mean (**b**), and absolute daily minimum/maximum (**c**). A larger circle size in **a** – **c** represents a location with an increasingly consistent diurnal cycle throughout the PI simulation.

Panels **d** – **f** show the dominating process (ARI/ACI) driving the total LWP response during the diurnal cycle (**d**), day (**e**), and night (**f**). A larger circle size in **d** – **f** represents a better match between the individual response (PD$_{ARI}$ or PD$_{ACI}$) and total response (PD). All panels show the AOD perturbation as contour lines at 0.05 increments.

**Point-by-point responses**

In the following pages we provide point-by-point responses to each comment. Reviewer comments are in blue. Author responses are in black and indented with new/revised text in red.
* * *
**Reviewer #1**

This study investigates the impacts of anthropogenic aerosols on radiation and cloud properties using a global convection-permitting model. Four 40-day simulations are conducted to isolate the aerosol impacts via the aerosol-cloud interaction and the aerosol-radiation interaction. The methodology and analysis are straightforward. However, several severe flaws in the manuscript prevent its publication in its current form.

Major comments:

Sections 2.2 and 2.3 are unclear. I don't understand what the authors changed for the three PD simulations. Only aerosol concentrations? Please provide more details.

> As suggested by the reviewer we have rewritten the two sections to provide more detail and clarification.

The authors mentioned deep convection throughout the manuscript but never provided any quantitative analysis about how much of the responses are for convection or large-scale environments. Even if the authors select several regions with frequent deep convection, I don't think convection is the only precipitating process in those regions. For example, how do you know how much precipitation is from deep convection?

> In our study we are primarily focused on the regional response, through which we can scale up to the global scale. The response of a single cloud is not necessarily representative of the regional-scale cloud field. Hence we are not trying to isolate every single deep convective core and attribute precipitation changes to it, but instead try and quantify the regional-response with the additional realism that the convection-permitting resolution provides. As we are focused on the regional scale it is difficult to quantitatively isolate the precipitation from convective cores and the larger-scale. Our model does not have an explicit distinction between convective (deep/shallow scheme) and large-scale precipitation that a typical GCM does.

> We use the diurnal composites to link the different responses together - through which we are able to highlight the role of deep convection in some regions, and large-scale ascent / subsidence in others. Deep convection is not the only precipitating process in the deep convective regions, but the diurnal cycle composites

clearly show a strong response coincident with the timing of deep convection, and this is supported by other variables.

In response to the reviewer's comment we have decomposed the 1-degree precipitation response for grids where there is ascent at 500 hPa. This provides some degree of separation between all grids and those that are more likely to have convection. The analysis (Fig. R2) shows strong consistency between the regional responses, consistent with our hypothesis that the precipitation response is occurring in the convective cores. For ARI in the Congo and the Amazon the magnitudes are equal, demonstrating that all the effects occur on ascending grid points. In the Maritime Continent, there is an additional impact on large-scale precipitation (persistent increase across the diurnal cycle) which is consistent with our analysis and reasoning. In Fig. R3 we additionally analyse the distribution of precipitation rates across each convective region using the 5-km resolution output. ARI and ACI impact precipitation across all intensities (Fig. R3 a – c) but the biggest impact to the total precipitation occurs at precipitation rates above 10 mm hr$^{-1}$ (Fig. R3 g – i), which are magnitudes associated with deeper convection rather than shallow convection. We include these two plots in the supporting information and refer to them in the revised manuscript.

[Figure]

Figure R2. Decomposed diurnal response of 1-degree precipitation in the convective regions for all grids (left column), and those with ascent at 500 hPa (right column).

[Figure]

Figure R3. Distribution of time series mean precipitation rates in each of the three convective regions. Rows show the percentage of grid cells ($\overline{N}_\%$) in each bin $d\log_{10}P$ (**a** – **c**), the total precipitation rate ($\overline{PN}_\%$) per bin (**d** – **f**), and the change in total precipitation rate ($\Delta\overline{PN}_\%$) per bin (**g** – **i**). Total precipitation rates ($\overline{PN}_\%$ and $\Delta\overline{PN}_\%$) for the Congo and Amazon are shown on the same scale to aid comparison.

The revised text is as follows:

The Congo and Amazon are strongly perturbed by anthropogenic aerosol from biomass burning sources. The primary driver is a localized modification to the convective environment that reduces daily accumulated precipitation by 1 mm day$^{-1}$ in both regions (~15 % and 10 % of PI values for Congo and Amazon). The precipitation response is associated with ascending regions and precipitation rates greater than ~10 mm hr$^{-1}$ (Figs. S6 and S7), linking the changes to convection.

The discussion of the results is verbose, while the reasoning is oversimplified throughout the manuscript. The latter severely degrades the quality of the study. Please explain your results more logically, not just provide some simple assumptions. Please find further details below.

We address this comment in several ways. We have removed the global-response section which, as pointed out by this and other reviewers, did not fit the focus of the study. A new section using the temporal decomposition tool (Section 3.1) expands the regional analysis to the full global coverage, which helps to refocus our study and improve the quality of the analysis and conclusions. We have gone through the manuscript and removed unnecessary passages of text and made the analysis more succinct. Where necessary we have provided additional reasoning or support to explanations and hypotheses.

Minor comments:

Line 7: Please provide the full name of "ICON" for its first appearance.

That has now been included.

Line 39: What do you mean by lack of memory?

'Lack of memory' refers to the tendency for convection on the regional scale to exhibit a memory effect (Colin et al., 2019). Past instances of convection can influence the initiation and development of future convection. We have changed 'lack of memory' to 'lack of convective memory' and cited Colin et al. (2019).

Line 43: "Limited area" to "Regional"?

This has been updated as suggested.

Line 44-45: What do you mean by large-scale controls on the availability of energy and water vapor?

The reviewer is referring to the lines:

"Limited area high-resolution models provide useful process insights (Marinescu et al., 2021), but their global representativeness and ability to respond to regional drivers remain unclear, and they do not satisfy large-scale controls on the availability of energy and water vapour (Dagan et al., 2022)."

Dagan et al. (2022) showed that the response of clouds to aerosol perturbations in regional-scale simulations is highly sensitive to the representation of the larger-scale thermodynamic environment (boundary conditions in a regional configuration). In some instances the idealized nature of the applied boundary conditions (e.g., time-independent forcings or a relaxation towards domain-mean properties) may end up controlling the regional-response, rather than the perturbation.

In response to the comment we have rewritten the sentence as:

..but their global representativeness and ability to respond to regional drivers remain unclear, and may not sufficiently represent the interaction between the large-scale thermodynamic environment and the regional scale (Dagan et al., 2022).

Line 62: What is the resolution of NICAM?

The NICAM resolution used in Sato et al. (2018) was 14 km. This has been included and the full name of NICAM has also been provided.

Line 66: "or" to "nor"?

Rewritten as suggested.

**Lines 74 and 75: The full names of ICON and MACv2-SP?**

Rewritten as suggested.

**Line 112: add "separately" after "cloud water."**

Rewritten as suggested.

**Line 115: What is SLEVE?**

SLEVE is the acronym given to a terrain-following coordinate formulation called the smooth level vertical (SLEVE) coordinate.

Whilst making the manuscript less verbose we have removed reference to this acronym and refer to the model description paper (Hohenegger et al., 2023) instead.

**Lines 140-141: Any references for such an adjustment?**

We will add references as suggested. The sentence now reads:

These figures are uncertain (Hamilton et al., 2018) and may substantially underestimate the anthropogenic contribution (Lauk and Erb, 2009). In our simulations, we enhance the anthropogenic component to 75%, which is consistent with higher estimates (Lauk and Erb, 2009, and references therein) and should provide a stronger signal in response to our perturbation

**Line 152: Figure 1 shows an overestimation of Nd. Do you have any statistical calculations validating the changes of aN and bN?**

The line in question is "This relationship provides more sensitivity than the original used in Stevens et al. (2017), but as we show in Fig. 1 results in a present day distribution of Nd consistent with observations."

The purpose of the experiment design is to examine and isolate the processes through which the aerosols impact the regional-scale climate. We have used a prescribed and idealized global aerosol perturbation in order to achieve this. The values of aN and bN were taken from a previous study that used in-situ values to make an empirical fit between AOD and in-cloud Nd. The authors used the parametrization to provide a robust perturbation in a study designed to focus on ARI and ACI-related processes. We follow a similar approach.

The spatial distribution and magnitude of AOD and Nd in the PD experiment are consistent with the ocean-limited coverage of observations (as shown in Figure 1) and provide a plausible perturbation. We also focused on the month of September to gain the largest global-mean perturbation. The idealized setup is therefore not

designed to provide a quantitative estimate of the anthropogenic aerosol forcing as in AeroCom style experiments, and we do not provide this.

A direct evaluation results in global means of 105 cm$^{-3}$ (observations) and 103 cm$^{-3}$ (model) and a correlation coefficient of 0.34. The poor correlation is primarily due to widespread overestimation of Nd over the SE Atlantic Ocean and the Indian Ocean. These are the regions that are directly influenced by the biomass burning aerosol and thus unsurprising that the comparison is weak. We also overestimate in the Pacific Ocean and underestimate close to the coastal regions of Europe and North America. This does demonstrate some weakness in the method, but is still reasonable given our focus on impacts to aerosol processes, rather than providing robust estimates of anthropogenic aerosol forcing.

In the revised manuscript we include the following text in Sect 2.3

A comparison between the simulated and observed $N_d$ (limited to the ocean) yields similar means (105 cm$^{-3}$ and 103 cm$^{-3}$) but a low correlation coefficient of 0.34. This is in part due to high simulated values over the biomass burning regions that are not reflected in the annual mean observations, but also due to regional variability that MACv2-SP does not capture. Despite the poor correlation, our idealized representation of aerosol provides appropriate perturbations to the radiative fluxes and bulk cloud properties that are spatially consistent with the dominant sources of global anthropogenic aerosol forcing.

The response to the following four comments has been combined

Section 2.2: The description of aerosols in ICON needs more detailed clarification. Which type of aerosol variables does ICON need? How do you consider pre-industrial and present-day aerosol conditions? The current description is confusing. Did you only add fN in the model? Where are Nd,cld and Nd,sfc from? Does ICON contain any aerosol microphysical processes (processes converting emissions to aerosol concentrations), or are aerosols entirely prescribed in the model (input aerosol concentrations, AOD, etc.)?

Lines 172-178: I am confused with what MACv2-SP provided to ICON. Anthropogenic aerosol concentrations?

Lines 179-180: Please rewrite this sentence.

Lines 179-185: Both PI and PD simulations contain biomass-burning sources. What are their differences? In Line 132, you mentioned that in MACv2-SP, biomass-burning emissions are anthropogenic sources. Could you please provide more details about that?

All of these comments refer to the aerosol representation (section 2.2) and coupling to the microphysics/radiation schemes (section 2.3). As suggested by the reviewer we have rewritten and expanded both sections to provide clarification / additional detail.

Lines 212-213: Can internal variability be eliminated by only focusing on regional responses?

The lines in question are

"The limited length of our simulations poses some issues as it makes it difficult to disentangle this internal variability from the global-scale responses to aerosol effects. Due to this, we will only briefly discuss the global scale responses, and instead focus on the regional responses due to aerosol, allowing us to mitigate or reduce the impact of the internal variability by compositing over multiple diurnal cycles."

The most prominent internal variability we observe is in the Southern Ocean (see Figure 3), where frontal features are often separated by 1000s km (e.g., Southeast Pacific) by the end of the simulation. However, over much of the subtropics and tropics, where our aerosol perturbations primarily are, the spatial heterogeneity occurs on much smaller scales (e.g., Southeast Atlantic Ocean). By focusing on regional domains there is a reasonable chance that we mitigate or reduce the impact of internal variability. Additionally, we composite the response onto single diurnal cycle, which increases the number of data points by a factor 30, further helping to reduce the 'noise' of internal variability (Noise $\approx 1/\sqrt{N}$).

In the current text we acknowledge that this method allows us to mitigate or reduce the impact, rather than eliminate it. This is demonstrated in a new figure (Figure R4; introduced in a comment below), which will be included in the supporting information. We also point out that the lines in question have been deleted along with the section.

Line 229-230: How did you know it is due to internal variability but not actual model differences? Internal variability refers to variability due to natural internal processes within the climate system. But here, you talked about the differences between the two simulations.

Line 234: Again, how did you know it is due to internal variability?

We believe these comments refer to lines 219-220 and 224. These lines have been removed (as part of Section 3) but as we discuss internal variability throughout the manuscript this comment is still valid and requires a full response.

The aerosol perturbation (the difference between the two simulations) will result in a redistribution of energy. Some of this will be coherent, e.g., increased cloud liquid water, but some will be random (e.g., a slightly misaligned front), and help to drive the spatial heterogeneity we observe. This redistribution may occur over seasonal/decadal timescales but in our case this is occurring over days. In both cases we would argue that it is reasonable to describe the redistribution as internal variability of the system, and is consistent with Schwarzwald et al. (2022):

"Internal variability consists of the naturally occurring variations in climate on timescales from daily weather to multidecadal processes due to interactions between various components of the Earth system. Internal variability is made

up of components that are predictable, such as the El Niño–Southern Oscillation, as well as irreducible uncertainty due to the chaotic nature of the system."

However, in the instances where the misaligned front or fractionally strengthened tropical cyclone is due to changes in the aerosol perturbation rather than internal variability our methodology will not be able to account for this. Hence, we only suggest internal variability as a likely candidate for the unaccounted responses in the temporal decomposition.

In response to the reviewer's comment we have clarified what we refer to as internal variability with the additional text below.

> "The limited length of our simulations poses some issues as it is difficult to disentangle internal variability from the global-scale responses to aerosol effects. By internal variability, we refer to the chaotic nature of the atmosphere, whereby small fluctuations grow rapidly in time. For example, Fig. 2 shows that as the simulation progresses the changes in aerosol concentration have large-scale impacts on the precise timing and location of atmospheric fronts, which appear as a regional change when differencing simulations, but are not usefully considered as a robust 'aerosol effect'. This behavior is similar to initial condition sensitivity where small-scale perturbations at the beginning of the simulation can quickly develop into pronounced changes (Keshtgar et al., 2023; Lorenz, 1963)"

Line 253: "internal variability" to "spatial heterogeneity." Please check the whole manuscript to ensure the correct terms are used.

See response above.

Line 235: add "outgoing" before "shortwave radiation."

The section and line in question has been removed, but we have clarified in further instances where appropriate.

Lines 269-271: Would applying the tool to Section 3 (original model data but not differences) be better also to remove your so-called "internal variability"?

We believe this is an excellent suggestion and we thank the reviewer. Following similar comments from the other two reviewers we chose to make major revisions to the manuscript. This is outlined below, with more detail provided at the beginning of the document.

We have replaced Section 3 (Global response) with a new section where we apply the decomposition tool globally, as the reviewer suggests. To achieve this we subset the globe into 15x15 degree regions, and applied the decomposition tool to each region. We apply thresholds to the response in each region to focus on regions where a robust response is observed, and we use the remaining data to examine the

role of ARI and ACI on the regional-scale, but for all regions of the globe simultaneously.

This major revision has greatly improved the focus and flow of the manuscript. To maintain this focus we have changed the title of he manuscript to "Variability of regional-scale aerosol impacts on clouds and radiation in global kilometre-scale simulations"

Lines 279-283: There is an assumption here: the long-term component is constant and independent of time. Is it correct? At least Figure 7f doesn't show that!

We assume that the long-term component of the decomposition may include a persistent (time-independent) response along with internal variability (time-dependent). A persistent time-independent response may be, for example, enhanced subsidence, an overall increase in LWP, or a persistent increase in cloud fraction.

The contrasting time series of the persistent response and internal variability is well captured by Figure 7f. In the long term response there is evidence of consistently enhanced LWP in the $PD_{ACI}$ and PD experiment, with a sharp decrease in the middle of the time series spanning 10 d. We attribute this sharp decrease to internal variability: All perturbation experiments share the same 10 d time series, strongly suggesting a synoptic feature in the PI experiment that was not present in any of the perturbation experiments. A similar feature can be seen towards the end of the long-term time series in Figure 7c (The Congo).

The introduction of the decomposition method is now in a new section (Sect. 2.4 Temporal decomposition of regional response) and has been rewritten to provide more clarity on how we isolate the persistent response from internal variability, and any assumptions and limitations. We have also included Figure R4 and associated text in the supporting information to further demonstrate the decomposition method.

[Figure]

**Figure R4**. Figure demonstrating the capability of the decomposition method. A synthetic response in (**a**) includes a short-term diurnal component with a time-dependent amplitude, a long-term persistent time-independent component, and internal variability from multiple sources. The aim of the decomposition method is to sufficiently isolate any underlying short-term response along with a persistent response. The result of the decomposition method from the manuscript applied to the synthetic response (short-term + long-term + internal variability[1-3]) is shown in (**b**).

Associated text in the revised supporting information:

Figure S1 demonstrates the ability of the temporal decomposition method to isolate short-term (diurnal) and long-term (persistent) responses amid internal variability. Multiple sources of synthetic internal variability / noise are added to a diurnal response with day-to-day variability in amplitude and a persistent (time-independent) response. This is shown in Fig. S1a. Our method aims to isolate the responses from the internal variability. Figure S1b shows the result of applying the decomposition method (Sect. 2.4). The method successfully isolates the responses. The total decomposed response is slightly higher than the synthetic response due to an imbalance in the synthetic internal variability from the third source. This demonstrates a weakness in the method when internal variability is strongly weighted towards a single direction of response (positive/ negative).

Line 286: Delete "with the SE Atlantic region".

This line was removed whilst making the manuscript less verbose.

Line 305: Do you have an explanation for the nighttime enhancement?

Response in question: ARI causes slight nighttime enhancement of LWP in the Southeast Atlantic.

There is a corresponding diurnal response in the liquid cloud fraction which drives the LWP response. The cloud fraction is enhanced due to an increase in latent heat flux from the surface to the atmosphere (~ -15 Wm$^{-2}$) which drives increased water vapour in the lowest 2 - 3 km. Additional heating due to the absorbing aerosol 'burns off' the cloud during the daytime, but as temperatures cool in the night the enhanced water vapour results in more cloud.

We have included the vertical profiles of LWC, IWC, W, theta, and Qv for the three non-convective regions in the supporting information. We also include additional decomposed variables of surface latent heat flux and liquid cloud cover. We refer to these in the revised manuscript.

Line 319: Does ACI increase cloud cover? Did you refer to Figure 4h?

Line in question:

"Over the Maritime Continent there is an additional enhancement of LWP throughout the diurnal cycle of +10%. This reflects the prevalence of low-level marine clouds over much of this region and may be partially driven by enhanced cloud cover."

ACI increases liquid cloud fraction by ~2.5% over the Maritime Continent. The cloud fraction response was originally included in Figure 9 but was removed prior to submission. The cloud fraction response over the Maritime Continent (and other regions) can now be found in new figures included in the supporting information.

The revised text reads:

Over the Maritime Continent there is an additional enhancement of LWP throughout the diurnal cycle of +10 %. This reflects the prevalence of low-level marine clouds over much of this region and is associated with enhanced cloud cover (Figs. 8b and S4) and evaporation from the ocean surface (Fig. S5).

Line 345: Please provide more details about how you constrain ascending air masses. Using hourly or daily data?

We constrain two variables to ascending air masses: $M_{flux}$ at 500 hPa and profiles of W. For $M_{flux}$ we remove all 1° data points where $W_{500hPa}$ is negative throughout the time series. A mean of the remaining data is taken at each timestep. For profiles of W↑ we have changed our method. Originally we constrained W to 1° grid points where $W_{500hPa}$ was positive. However, when there is a decrease in the number of updrafts within a region (as we hypothesize with ARI) the mean profiles of W↑ from PD and PI are not directly comparable, and of less use to the analysis. In our revised method we constrain W to the PI grid points where deep convection is likely to be occurring. This is achieved by taking a time series mean $W_{300hPa}$, which coincides with the altitude of maximum vertical velocity. Any grids with a positive $W_{300hPa}$ (ascent) are used to subset the PD experiments. We have renamed this variable W*.

We did not describe the remapping method previously, and so include a description of this in the revised methodology. By regridding the data we have already smoothed some of the ascending regions, so it is more appropriate to refer to this as ascending regions, rather than individual updraughts. We provide more detail on this and the constraining method as suggested by the reviewer in several locations:

In Section 2.4:

Data from the 5 km grid is regridded onto a regular 1° grid using the Climate Data Operators (CDO; http://www.idris.fr/media/ada/cdo.pdf, last access: 20 January 2025) software operator *gencon* which generates first order conservative remapping weights. As we focus on regional responses, we do not lose any information through the re-gridding process.

In Section 3.1:

Figure 7 shows the local solar time (LST) of maximum response during the diurnal cycle for LWP, ice water path (IWP), precipitation, and the cloud condensate mass flux at 500 hPa ($M_{flux}$) constrained to grid points throughout times series where $W_{500hPa}$ is positive.

In Section 3.2.2

Vertical velocity profiles are calculated on 1° grid boxes where the mean $W_{300hPa}$ across the PI time series is positive.

Line 349: Why not decompose them given that you did it in Section 4.1?

We were unable to decompose the profiles as output on all pressure levels was 3-hourly, which is insufficient for the decomposition tool. We have clarified this in the revised manuscript:

The frequency of output on all vertical levels is insufficient (3 hr) to robustly decompose the time series following Sect. 2.4, hence the profiles include influence from internal variability. To minimise this, the responses are composited onto a single diurnal cycle. Additionally, the limited day-to-day variability evident in Fig. 10 for the regions provides confidence that the responses are primarily due to the aerosol perturbation.

Line 357: Does the profile of potential temperature changes align with the anthropogenic aerosol loading profile? Do you have any model output showing the suppression of boundary layer mixing and drying aloft? How do you define the boundary layer? 1 km? 2 km?

The aerosol profiles are consistent with the region of maximum increase in theta (the lowest 4 km). We do not define the BL nor estimate its height, and agree that this is misleading. The profile responses are consistent with suppressed mixing in the lower atmosphere, and a reduction in Qv between 3 and 5 km demonstrates drying aloft.

We have rewritten this sentence for clarity and we have included profiles of the aerosol optical thickness perturbation with each of the variables.

Line 360-362: Did you mean the daily mean or afternoon mean? If you meant afternoon mean, Figures 11a and 11d showed that Amazon has weaker w effects than Congo. If you meant daily mean, Figure 8 shows the ARI effect is the most significant in the afternoon. Please provide more reasonable descriptions and explanations for the results.

The magnitude of W is greater over the Congo than over the Amazon. Hence the same absolute decrease in W equates to a stronger perturbation in the Amazon when compared to the Congo. This should have been clearer. We have rewritten this sentence to improve clarity.

This is consistent with the findings of Herbert et al. (2021). However, the percentage response of the W* profile and LWP is greater in the Amazon and suggests the

differences may be due to stronger capacity to buffer the perturbation over the Congo, which tends to exhibit more convection than the Amazon, or differences in the convective environments (Storer et al., 2010).

Line 367: How does the delayed release of CAPE increase high-altitude IWC?

The absorbing aerosol stabilises the thermodynamic profile via ARI (Wang et al., 2013), yet CAPE is still produced. Herbert et al. (2021) found that over the Amazon the CAPE was released later in the afternoon, driving some convection yet not to the full extent as without the presence of aerosols. Relative to the control simulation, IWP was suppressed during the early afternoon but enhanced later in the evening when the delayed convection occurred. This is consistent with our results: IWP is strongly suppressed during the early afternoon and weakly enhanced during the evening.

We have rewritten this statement to clearly describe our reasoning.

Changes to convection and the vertical transport of condensate strongly suppresses IWP during the afternoon, with a smaller enhancement during the evening. This is consistent with Herbert et al. (2021) who found that absorbing aerosols over the Amazon caused the accumulation of convective available potential energy (CAPE) to be released later in the afternoon, driving some convection yet not to the full extent as without the presence of aerosols.

Line 367-370: How did you know convection occurred certainly given increased CAPE aloft? More evidence is necessary to support such types of conclusions.

We have removed this sentence.

Lines 380-381: Which variables did you refer to?
Lines 381-383: Please provide more details about the reasoning.
Line 391: What do you mean by delayed CAPE? In addition, how can they explain the non-linear response?

The paragraphs that include lines 380 to 391 have been rewritten to a) refocus the points being made, and b) make the manuscript less verbose. The new paragraph reads:

The Congo and Amazon regions respond consistently to ARI and ACI individually, but the total responses are different, suggesting a degree of state dependence. In the Congo region (Fig. 11) the responses of many variables have largely additive contributions from ARI and ACI (e.g. θ, W*, IWC, LWC above 2 km, and precipitation), with ARI tending to drive stronger regional responses than ACI. In contrast, the Amazon region (Fig. 12) does not consistently show additive responses. Some variables show largely additive responses (e.g. LWP, precipitation, W* below 7 km) driven primarily by ARI but others are not clearly attributable to either ARI or ACI (e.g. IWC, W* and θ above 7km). In contrast to the Congo region, ACI plays a stronger role in the Amazon and is responsible for most of the LWP response and

LWC, but only weakly impacts precipitation. The enhanced role of ACI in the Amazon is consistent with relatively higher frequency of shallow convection than in the Congo, which is a regime known to be sensitive to ACI (Langton et al., 2021; Sheffield et al., 2015). The contrasting roles of ACI and ARI in the Congo and the Amazon suggests the response of convective environments to changes in the aerosol population is dependent on the background state, which has also been observed in remote-sensing studies of the Amazon region (Herbert and Stier, 2023; Ten Hoeve et al., 2011; Yu et al., 2007). This is consistent with Chang et al. (2015) who show that the response of deep convective clouds to aerosols is regime-dependent due to non-linearity between dynamical and microphysical processes

Line 400: Is 1 mm day-1 consistent with Figure 9?

Yes, this is consistent.

Line 415: Do you mean the red line? It is over zero at all hours.
Lines 416-417: ARI is positive in Figure 9h.

The total response includes a persistent response and a diurnal response. The diurnal response will always vary about zero, so any systematic increase or decrease is due to the persistent response. In the line referred to by the reviewer we discuss the changes "relative to the persistent response". For the variable in question the persistent response is large enough that the total response remains above zero throughout the diurnal cycle.

We recognize that this is insufficient information for the reader. The sentence has been rewritten to improve the clarity.

Lines 417-418: The ACI effect on LWP is always positive, while its effect on Mflux is negative only between 12:00 and 15:00 LST. How can you explain it?

The mass flux response is a function of the change in vertical velocity and the condensate loading. Therefore the LWP and Mflux response will not necessarily be in the same direction. In this instance the reduction in afternoon Mflux amidst an increase in LWP suggests some afternoon suppression of W. This is consistent with the Maritime Continent region as a whole which includes convection over land and low-altitude marine clouds.

Lines 422-428: Please provide more detailed reasoning!

We have amended the sentences as suggested.

Line 440-441: ACI and ARI are comparable in the ice-phase region!

The reviewer is correct. Although ARI clearly dominates the response in the warm-phase region, neither pathway dominates in the ice-phase region, and there is a suggestion that both play a role in the total response. This has been rewritten.

**Reviewer #2**

The authors take a crude aerosol parameterization (MACv2) and proceeded to parameterize it further for use in km-scale aerosol sensitivity studies. It's still crude, though it seems to be more intelligently refined. Overall, the authors try to present these one-month-long km-scale simulation as a way to study generic features of aerosol radiative response. I don't think they made a convincing case. As Reviewer 1 indicated, there are "several severe flaws" in this work. However, I would go further than Reviewer 1 to say the methodology is also flawed in that it is not fit for the purpose at hand. I don't think this manuscript should be published in ACP in its current form, and I do not think it is likely any revision along the lines of the submitted manuscript will make it closer to acceptable form. Besides the points raised by Referee 1, here are additional points that will likely prevent this manuscript from proceeding further.

We thank the reviewer for their comments and criticisms, but we strongly disagree with all points raised in this summary.

A significant criticism is in the methodology of our study. Regional-scale simulations have been prolifically used in the community to focus on the impact of aerosols on clouds/precipitation/radiation in order to improve our process-level understanding. These studies frequently use prescribed aerosol fields in order to focus the study and remove a source of considerable uncertainty due to aerosol-related processes. They are typically simulated on timescales of a single day to weeks.

Our study is well aligned with these studies. In our study we perform the same type of simulation but across the entire globe for 40 days, which in itself is an uncommonly long timescale. With this design we can explore the role of ARI and ACI, directly akin to a regional convection-permitting simulation, but simultaneously across the entire globe, encompassing different convective environments, thermodynamic environments, large-scale drivers, aerosol-types, cloud distributions etc. This provides an unparalleled opportunity to identify consistent, generic, responses in different environments. This can be achieved to a certain extent by collating regional simulation studies from across the literature, but will inherently include significant uncertainty due to the different models, configurations, assumptions, microphysics etc used in each study. The regional simulation studies may also include deficiencies from predefined boundary conditions that may erroneously influence the response (e.g., Dagan et al., 2022) - something that our methodology largely overcomes.

Alternative studies have been designed to provide quantitative estimates on the radiative forcing of anthropogenic aerosols, or the long-term response of the hydrological cycle. In these simulations, such as those in the AeroCom initiative, global aerosol microphysics schemes are coupled to the atmospheric component and run for decadal timescales. This type of experiment is required to robustly estimate

the global radiative forcing of anthropogenic aerosol, but is not well-suited for exploring process-level understanding (ARI/ACI). An AeroCom style experiment is not what we are attempting to do in this study, though the reviewer's comments suggest this is the direction they believe we are aiming.

This misunderstanding may partly arise from the Global response section, in which we discuss global-mean responses of the aerosol during the time series. This analysis was primarily used to demonstrate that the idealized nature of the experiment design and the simulated responses were consistent with what we may expect. However, it is likely that this has detracted from the core of the study.

In response to this potential misunderstanding, and comments from the other reviewers, we have removed the global response section and instead we present additional analysis that keeps the focus of the study on the novel aspect of the study. We have also changed the title to "Variability of regional-scale aerosol impacts on clouds and radiation in global kilometre-scale simulations". Our results are consistent with other regional-scale studies with similar focus but we are able to extend to the whole globe. We describe the changes in more detail at the beginning of this document.

The authors should release the code for the model itself, the code edits required for their specific MACv2 setup, and the resulting MACv2 files (Nd, optics, etc.) as part of this manuscript. The analysis code released is not sufficient for reproducibility (anyone can make up a few netcdf files). I understand there may be limitation at some European centers regarding data/code transparency, but there are workarounds. If not possible, detailed explanation must be given why it is not possible. Note that this is not a request to release model outputs (though that can be nice; and the authors should definitely consider releasing those too), it is simply the underlying code/files that should be made public (as much as possible).

As requested by the reviewer we will provide all necessary information.

The model we used is publicly available and released under the permissive BSD-3C licence and can be downloaded here: https://gitlab.dkrz.de/icon/icon-model. This includes the implementation of MACv2-SP, which is also publicly available as supplementary information in Stevens et al. (2017). Full access to the data can be granted (and processing resources) for anybody interested via the DYAMOND initiative website (https://www.gewex.org/dyamond/).

We include this information in the Code and data availability section:

The ICON model is available under an open source (BSD-3C) licence (https://www.icon-model.org) and publicly available at https://gitlab.dkrz.de/icon/icon-model. The MACv2-SP software is implemented in the ICON model source code, and is publicly available in the supplementary material of Stevens et al. (2017). Full access to the simulation output data and necessary

processing resources is available upon request via the DYAMOND initiative website (https://www.gewex.org/dyamond/).

The design of the simulations (namely 40-day DYAMOND cases) is not really appropriate for the aerosol response being targeted here. The authors make note of this serious issue several times in the manuscript (see below), but they somehow overcome it without much explanation, or maybe I missed it. Why do you think you say so much about ARI and ACI in 30-day runs? That doesn't seem quite right to me. There's simply too much noise (internal variability, etc.) in this for any result to be meaningful.

This comment is echoed in the reviewer's summary and was addressed in our response.

More clarity about the one-way coupling here will be beneficial, and in so doing, it is important to highlight how limiting the setup is. By one-way coupling, I mean that MACv2 prescription affects the optics, radiation droplet number, and cloud droplet number, but nothing in the model affects the MACv2 prescription. Is that correct? If so, I think it should be highlighted more prominently — sections where we cannot say much definitively about what's going on should be deleted (e.g., S 3). As a corollary, do you think the one-way coupling will make the response overestimated or underestimated?

We have expanded Section 2 to provide more detail and clarity on the manner in which the aerosol is coupled to the model. Section 3 has now been deleted and we include a discussion of this limitation using previous studies to hypothesise how an interactive representation of aerosol would impact our results.

Potentially, the authors can consider submitting aspects of this work to GMD (e.g., focusing on the modifications to MACv2 and maybe some aspects of remapping, etc.). For ACP, I think a refocused, less verbose manuscript can potentially be useful for the community. However, the manuscript should narrowly tackle what is possible and avoid less certain topics.

The manuscript is not well-suited for GMD as we are not presenting a model development, but instead using an existing model with minor additions to study aerosol-cloud-radiation interactions. This study is well aligned with the scope of ACP.

Changes to Section 2 will help clarify the modifications that have been made to the existing model framework (ICON coupled to MACv2-SP), and the major revisions we have made (see beginning of document) have ensured the manuscript has a clear focus.

Below are some comments I wrote down while (re)reading the manuscript:

L 23 and elsewhere: I would talk about ARI before ACI (because that's the logical progression, think direct vs indirect).

Where appropriate we have done as suggested.

L 29–24: Hmm, is that really "a primary" one?

We have rephrased this sentence.

L 73: What's "well defined" here? And why do you feel the need to say so? Are you trying to say people before this study used poorly defined treatments?

We have rephrased this sentence.

L 76–80: the manuscript is already tortuously long; these types of meaningless lines can be deleted.

As suggested by the reviewer this has been deleted. We have also gone through the manuscript and attempted to minimise similarly redundant paragraphs etc.

L 86: This is not a classic "AMIP" experiment by any stretch of definition, or am I confused? (Cf. L 121 and thereabouts.)

It is similar to AMIP experiment protocols in that the sea surface temperature and ice content are fixed to climatologies, and therefore it is an atmosphere only configuration. We refer to "AMIP" on L86 in reference to the SST/SIC climatologies that we use.

Fig 3 and associated text: What's the deal with the "upscaled" panel? Are you simply remapping the middle panel to 2-degree resolution? If so, and if you really want to discuss this, you will have to give more details about the remapping algorithm and all sorts of things associated with this. I don't quite see the point of all of this though, so more motivation may be needed to begin with…

We have removed the third panel as suggested.

L 210–215: I don't think this is enough to circumvent these serious issues. Can you give more reasoning why you think you'd adequately address these challenges by doing something different?

In response to this comment and others we have removed this section and analysis of the global scale mean response.

L 234–235: Yeah, or at least, we simply don't know…

In response to this comment and others we have removed this section and analysis of the global scale mean response.

S 4: Why do we need a global model to study regional responses? Maybe some regionally refined setup will be more useful (much cheaper) than we have here?

Regional simulations are cheaper to run, but only provide information on one region. Hence we can only quantify the role of aerosols on the regional-scale climate in one location. These may not be applicable to other regions due to different meteorological conditions and non-linearity, therefore the regional studies need to be performed for all regions where we anticipate a role of aerosols.

As demonstrated by Marinescu et al. (2021) and others, the response of clouds to aerosol is sensitive to the model and the microphysics. Hence, relying on singular regional simulations (from different models) to build up our understanding of global responses to anthropogenic aerosol will inherently include similar uncertainty. Using one model to simultaneously focus on aerosol-cloud-radiation interactions across all regions of Earth removes a lot of this uncertainty and allows us to identify consistent processes/drivers/pathways across the globe. Ideally this would be repeated using other models, and is the focus of future DYAMOND style experiments, which we discuss in the conclusions.

Aerosols have local and non-local effects. A limited-area regional simulation cannot provide information on the non-local response, whereas a global model can. We need the spatial-scale of a global simulation but the ability to sufficiently represent the key processes, including those at the cloud-scale. This configuration is able to satisfy both. Though we cannot fully exploit this in our simulations, we hope that ours can inform the configuration and design of future experiments.

L 450: I cannot find this reference; looks like it has not been published yet? If so, maybe this is an improper citation…

We apologize for this. We can confirm that this study is now published and the reference is legitimate and proper.

L 454: I don't think we can say that ("link"), due to all sorts of challenges (internal variability, etc.) related to the design.

The model design allows us to focus on the underlying pathways and processes (links) through which the aerosol impacts the cloud droplet number concentration and the radiative properties of clouds and the atmosphere.

This has been rewritten as

This allows us to identify pathways and key processes through which the aerosol perturbations impact radiative fluxes and cloud droplet number concentration.

L 466: Future direction for…?

This references "Upscaling the analysis to a coarser resolution of 2 degree highlights the recovery of some features (Fig. 3) and a possible future direction"

This has been removed, along with the panel as suggested by the reviewer in a previous comment.

L 466–467: I am not following — perhaps elaborate further here and elsewhere?

This paragraph that contained these lines has been removed.
* * *
**Reviewer #3**

Review of egusphere-2024-1689: Isolating aerosol-climate interactions in global kilometre-scale simulations

Summary: In this study, the authors utilized month-long, global simulations with 5 km horizontal grid spacings to assess the aerosol-radiative impact, the aerosol-cloud impact, and their combined effects, using current day and pre-industrial aerosol estimates. Aerosols are introduced into their model by affecting the number of cloud droplets, the cloud droplet effective radius, and through AOD impacts on radiation. While I commend the authors on running these computationally expensive, global simulations, I question whether their framework and model can be used to come to the process-level conclusions that are being drawn in this study. Furthermore, some of the methods and model set-up can be made clearer. Throughout their results, the authors provide plausible explanations for their results, but they often seemed to be speculative, as opposed to rooted in analysis from the model data. As such, while global, long-term, simulations of aerosol effects do provide unique science opportunities (such as impacts on the larger-scale features that can be better resolved with 5 km grid spacing), this manuscript seems to focus on more uncertain and less justifiable aspects of their simulations.

We thank the reviewer for their comments. We have made revisions to the manuscript to remove the more speculative elements of our manuscript. This includes the global-mean response section (Section 3) which has now been replaced with a global analysis of the responses using regional-scale decompositions as suggested by this reviewer in a later comment. In the remaining results sections we have revised the manuscript to make sure our explanations are supported by either the figures in the manuscript, additional figures in the supporting information, or from the analysis/conclusions of previous studies. Any explanations that are not fully supported have either been removed or noted as such.

Major Comments and Concerns:

The authors state that "the novel aspect of this study is the globally resolved deep convection" (L262). However, 5 km grid spacing does not resolve most deep convection (i.e., isolated and scattered deep convective clouds), so the authors should reconsider framing their study in this way. This is particularly concerning given that the authors choose their regions of analysis due to having deep convection. This may require a shift to better resolved features in their simulations.

A numerical model of the atmosphere will resolve convection on the scales that are represented in the model, if not, parameterization for the unresolved convection is

active. At very coarse resolutions convection will be under-resolved and some aspects of convection will not be perfectly represented. We agree that a horizontal resolution of 1 km would be preferable, but at 5 km resolution many aspects of convection are already better represented than in a model with parameterized convection. Among those better represented aspects are for example the diurnal cycle of convection and the coupling of convection with its environment (important for this study), which is especially true in tropical regions dominated by large convective clusters. There is a large body of literature demonstrating this, which goes back at least two decades, especially in the area of limited area modelling. We try to avoid making "the perfect the enemy of the good" here. A recent study describing the simulation of convection with ICON in a comparable configuration to our setup was conducted by Segura et al. (2022).

We have added the following text to the methods section (Sect. 2.1)

In our experiments, we use a horizontal resolution of approximately 5 km with 90 levels from the surface to 75 km corresponding to a vertical resolution of about 25 to 400 m (Hohenegger et al., 2023, G_AO_5km setting). This configuration of ICON does not explicitly resolve the smallest scales of convection (< 5 km) but has been shown to reproduce observed diurnal and seasonal cycles of tropical precipitation Segura et al. (2022). Given that ESMs tend to use spatial resolutions of tens to hundreds of kilometres, this makes a marked improvement in our ability to resolve many aspects of convection (Done et al., 2004; Prein et al., 2013) and is well-suited to our study.

And the following to the conclusions section:

Additional sources of uncertainty arise from the cloud microphysics scheme and unresolved convection. The choice of the cloud microphysics scheme and representation of cold-phase processes has been shown to impact the response of convective clouds to aerosol (Heikenfeld et al., 2019; White et al., 2017; Sullivan and Voigt, 2021; Marinescu et al., 2021). Archer-Nicholls et al. (2016) and Possner et al. (2016) have shown that the magnitude of ACI and ARI responses may be sensitive to unresolved convection at 5 km resolution, potentially requiring a finer global resolution (e.g. Wedi et al. (2020)).

Many studies have shown that aerosols have a clear diurnal cycle. Given the focus of this study's analysis on the diurnal cycle of aerosol effects, can the authors comment on and justify their use of a time-independent aerosol perturbation that does not vary diurnally? This simplification seems especially concerning, given the focus on the diurnal cycle of aerosol effects in this study.

We agree that studies demonstrate a diurnal cycle of aerosols, but in many instances there is no consistent diurnal cycle. For instance, Yu et al. (2021) demonstrate that the diurnal cycle of dust from five globally-important source regions is not spatially nor seasonally consistent. This occurs due to the regional-dependent drivers of dust emissions, including convection, windspeed, and wildfires. Variability in the diurnal

nature of aerosols from other species is also demonstrated by Torres and Ahn (2024). A lot of the diurnal features of aerosol emissions and concentrations are driven by small-scale features of the atmosphere and surface, which will themselves be prone to considerable uncertainty and likely poorly represented in aerosol microphysics models. This is the uncertainty we try to remove from our study by using prescribed aerosol fields, however, we recognise the feedbacks between in-situ aerosols and their emissions would be a particularly important response. To achieve this we need interactive aerosols in global high-resolution models, which will be the focus of future studies (e.g, Weiss et al., 2024).

For these reasons we disagree with the comment that the simplification is especially concerning. However, we agree that the impact of, and feedbacks to, the diurnal cycle of aerosols is an important consideration. We have acknowledged this limitation in the conclusions section as follows:

The use of non-interactive and spatially invariable aerosol may be masking important feedbacks and processes. This includes the strong impact of clouds and precipitation on the spatio-temporal distribution of aerosols, changes to the surface properties, fluxes and turbulence that would influence emissions and aerosol removal processes. Aerosol emissions also exhibit diurnal cycles (Yu et al., 2021; Torres et al., 2024) that we do not account for. This gap will be closed by future aerosol perturbation experiments in global kilometre-scale models with an interactive treatment of aerosols and clouds.

L211-215: The authors state that they will only discuss the global scale briefly due to internal variability but can mitigate internal variability on a regional scale by compositing over multiple diurnal cycles. It is unclear why this same method cannot be applied globally, and whether this compositing can be used to reduce internal variability in these simulations. Internal variability comes up several more times in the manuscript, so being clearer about internal variability, and its impact on this study up front would be helpful.

We agree with the reviewer and thank them for this excellent suggestion. In response to this, and similar comments from the other reviewers, we have made major revisions to the manuscript following the suggestion made here. A more detailed description can be found at the beginning of the document.

With respect to internal variability, we discuss this more in the revised manuscript, and we include a new figure in the supporting information (Figure R4 in the response to reviewer #1) that helps to demonstrate how the decomposition method helps reduce or mitigate the effects of internal variability in the response.

Aerosol-Radiation details. It seems that plumes of biomass burning emissions and industrial emissions are used. Do these aerosols have different radiative properties that are interacting with the radiative scheme? Where do the aerosol particles live in the vertical? Are they advected over the course of the simulation? The authors state that the nine plumes are configured to reproduce the AOD for the year 2005 (L134), but how are they configured? I think this section should have more details, which would improve the clarity of the experimental set-up.

We have rewritten Sections 2.2 and 2.3 to provide more detail and clarity on how the aerosol is represented in the model. We also include the resulting vertical profiles of aerosol extinction from the six regions in the supporting information.

There have been countless limited-aera modeling studies of aerosol effects, specifically ones that focus on the regions highlighted in this study. However, there were only a few limited-area studies included in the authors' introduction or through their results section. Process-level insights from additional limited area studies may help the authors disentangle and contextualize their results.

We have included additional references throughout the revised manuscript.

Minor comments:

L4: "we have been unable to explicitly simulate cloud dynamics." We have been able to in limited-area model studies, so can you please make this more explicit about global models.

This has been rewritten

Due to computational constraints, we have been unable to explicitly simulate cloud dynamics in global-scale simulations, leaving key processes, such as convective updrafts, parameterized.

L44: What does "regional drivers" mean here?

Different regions will have different drivers that influence the thermodynamic environment, or diurnal cycle. This means the role of aerosols in one region will not necessarily translate to other regions. We have rewritten this sentence for clarity.

L97: It is unclear to me why the same number concentration cannot be used for both radiation and microphysics? This seems like an unnecessary source of inconsistency.

The Nd concentrations are used in the radiation scheme and microphysics scheme but in our version of ICON the two are not coupled. Therefore the treatment of Nd in each scheme has been developed independently. In the radiation scheme Nd influences the cloud droplet effective radius, and when developed, was tuned to produce observationally-consistent cloud radiative properties. Similarly, in the microphysics scheme, Nd influences the warm-rain process via the autoconversion rate. When developed, Nd values were used to produce realistic precipitation rates. In an ideal scenario the two schemes would be coupled, and this is the focus of future development work for ICON.

For these reasons, we intended to make only minor modifications to the existing model, and maintain the treatment of Nd in each scheme. Larger modifications, specifically to the radiation scheme, may have resulted in significant changes to the global energy budget, resulting in an unrepresentative model. We have attempted to make this clearer in the revised Sections 2.2 and 2.3.

L134: Unclear how the extrapolation is done here. Can you make clearer?

This has been expanded to provide more detail.

L197-202: The authors state several important features are well-simulated in their simulation, but it is difficult to see from Figure 2 whether these features are being simulated properly.

This figure and accompanying text has been removed in order to focus on the new global-scale analysis.

L215-225: Most of this section focuses on results from the spin-up phase. Why do the authors focus so much on the spin-up phase, as this phase is used to spin up their model?

This section has been removed in order to focus on the new global-scale analysis.

L267-272: The authors separate their time series into short-term and long-term components, which reduces contributions from internal variability but introduces "non-local impacts to the region." More details are needed here in terms of how this reduces internal variability, and what is meant by non-local impacts.

L275: The "second application of the decomposition tool with a large prescribed periodicity." What is the prescribed periodicity, and how is this used to provide new, helpful information? More details would be helpful here.

In response to these two comments we have expanded our description and explanation of the decomposition method. We also include a new figure in the supporting information (see Figure R4 in the response to reviewer #1) that helps demonstrate how the method is able to reduce the impact of internal variability.

L349: Given the importance of removing internal variability, why aren't Figures 10 and 11 decomposed with similar methods as the other figures?

We were unable to decompose the profiles as output on all pressure levels was 3-hourly, which is insufficient for the decomposition tool. We have clarified this in the revised manuscript:

The frequency of output on all vertical levels is insufficient (3 hr) to robustly decompose the time series following Sect. 2.4, hence the profiles include influence from internal variability. To minimise this, the responses are composited onto a single diurnal cycle. Additionally, the limited day-to-day variability evident in Fig. 10 for the regions provides confidence that the responses are primarily due to the aerosol perturbation.

L391: The authors mentioned delayed CAPE as a potential process? Did the authors look at this in the simulations or is this speculation?

This hypothesis is based on a previous study (Herbert et al., 2021) that focused on aerosol impacts to deep convection. We do not calculate values of CAPE and we do not look further into the non-linearity. This sentence has now been removed to refocus the paragraph.

Figures 8-11: Given that the author's describe the results for each region at a time, it may be easier for the reader if the authors combined these figures based on region. I found it challenging to jump around from each figure to understand how these different variables come together to tell a consistent story.

We agree that this would help the reader navigate the figures. We have combined the figures and now show all variables (and corresponding response) for each region individually. We have also included the vertical profiles of aerosol optical thickness perturbations to panels that show the response of a vertical profile.

[Figure]

FIG R5. Example of the new 'combined' figure as suggested by Reviewer 3.

**References:**

Colin, M. and co-authors: Identifying the Sources of Convective Memory in Cloud-Resolving Simulations. *J. Atmos. Sci.*, **76**, 947–962, https://doi.org/10.1175/JAS-D-18-0036.1, 2019.

Dagan, G. and co-authors: Boundary conditions representation can determine simulated aerosol effects on convective cloud fields, Communications Earth & Environment, 3, 71, https://doi.org/10.1038/s43247-022-00399-5, 2022.

Done, J., and co-authors: The next generation of NWP: explicit forecasts of convection using the weather research and forecasting (WRF) model, Atmospheric Science Letters, 5, 110–117, https://doi.org/https://doi.org/10.1002/asl.72, 2004.

Grosvenor, D. P. and co-authors: Remote Sensing of Droplet Number Concentration in Warm Clouds: A Review of the Current State of Knowledge and Perspectives, Reviews of Geophysics, 56, 409–453, https://onlinelibrary.wiley.com/doi/pdf/10.1029/2017RG000593, 2018.

Hohenegger, C., and co-authors: ICON-Sapphire: simulating the components of the Earth system and their interactions at kilometer and subkilometer scales, Geoscientific Model Development, 16, 779–811, https://doi.org/10.5194/gmd-16-779-2023, 2023.

Marinescu, P. J., and co-authors: Impacts of Varying Concentrations of Cloud Condensation Nuclei on Deep Convective Cloud Updrafts—A Multimodel Assessment, Journal of the Atmospheric Sciences, 78, 1147–1172, https://doi.org/10.1175/JAS-D-20-0200.1, 2021.

Prein, A. F., and co-authors: Added value of convection permitting seasonal simulations, Climate Dynamics, 41, 2655–2677, https://doi.org/10.1007/s00382-013-1744-6, 2013.

Schwarzwald, K. Lenssen, N., The importance of internal climate variability in climate impact projections, Proc. Natl. Acad. Sci. U.S.A. 119 (42), https://doi.org/10.1073/pnas.2208095119, 2022.

Segura, H., and co-authors: Seasonal and Diurnal Features of Tropical Precipitation in a Global-Coupled Storm-Resolving Model, Geophysical Research Letters, 49, https://doi.org/https://doi.org/10.1029/2022GL101796, 2022.

Stevens, B., and co-authors: MACv2-SP: a parameterization of anthropogenic aerosol optical properties and an associated Twomey effect for use in CMIP6, Geoscientific Model Development, 10, 433–452, https://doi.org/10.5194/gmd-10-433-2017, 2017.

Torres, O., Ahn, C.: Local and regional diurnal variability of aerosol properties retrieved by DSCOVR/EPIC UV algorithm. Journal of Geophysical Research: Atmospheres, 129, e2023JD039908. https://doi.org/10.1029/2023JD039908, 2024.

Wang, Y. and co-authors: New Directions: Light absorbing aerosols and their atmospheric impacts, Atmos. Environ., 81, 713-715, https://doi.org/10.1016/j.atmosenv.2013.09.034, 2013.

Weiss, P. and co-authors: ICON-HAM-lite: simulating the Earth system with interactive aerosols at kilometer scales, EGUsphere [preprint], https://doi.org/10.5194/egusphere-2024-3325, 2024.

Yu, Y., and co-authors: A global analysis of diurnal variability in dust and dust mixture using CATS observations, Atmos. Chem. Phys., 21, 1427–1447, https://doi.org/10.5194/acp-21-1427-2021, 2021

---

## Author Response (AR2)

We thank both reviewers for their comments. We address each in turn below, with responses indented, and revised text in red.
* * *
**Reviewer 1**

**I think this is a significantly improved manuscript. It's very close to publication quality. I think this manuscript deserves to be highlighted, and I thus I reviewed it below as a highlights candidate. In my opinion, it requires a bit of editing to make it highlights worthy, which I hope the authors agree with. I list my comments below for potential improvements, and I hope the authors find them constructive. All my comments below should be considered minor (unless the authors deem them otherwise) and they should be considered non-blocking for publication.**

> We are delighted by this summary and very pleased that our revisions have addressed the reviewer's concerns and helped to improve the manuscript. We agree with all of the suggestions that the reviewer has provided and have made appropriate revisions. We hope that the revised manuscript has been enhanced to the level required to be considered a highlights candidate, as suggested by the reviewer.

**Minor comments:**

**The manuscript is still hard to read; if the authors can spare some time improving readability and presentation, that will go a long way. In general, it's improved significantly, and it is publishable as-is, but it could be improved and should ideally be highlighted (because it is a pretty good study)**

> We have made changes throughout the manuscript to improve the readability. This includes revisions in response to the reviewer comments and additional revisions.

**There several minor/technical issues that may warrant minor fixes (I list all of these below). I encourage the authors to fix them**

**I think the authors downplay the usefulness of their idealized setup, MACv2-SP. I think it would be nice if the authors make it clear where they see advantages in simplified schemes beyond computational performance (e.g., causality, process isolation) and where they see shortcomings (e.g., non-interactive nature of aerosols and lack of mesoscale features shown in ICON-HAM-Lite). The authors do plenty of the latter (shortcomings) but not enough of the former (advantages). After all, you're using a simplified scheme and if the only reason you're doing this study is because you had to, then that's no good. I think there's value in these simplified schemes in that they allow us easier access to some process aspects that are significantly harder to disentangle with fully interactive schemes. If you disagree with me, feel free to ignore. I just feel it is a missed opportunity not to forcefully defend these simplified schemes as practical and appropriate for some endeavors (you could also make your argument by carefully citing Stevens et al and Fiedler et al papers where they described the original MACv2-SP scheme)**

We agree with the reviewer and have included a new paragraph in the introduction that outlines the benefits of using an idealized representation of aerosols. The new paragraph is below.

*"Aerosols themselves are also a source of uncertainty in ESMs and high-resolution simulations due to complex aerosol microphysical processes that are poorly constrained or inadequately represented (White et al., 2017; Sand et al., 2021; Vogel et al., 2022; Regayre et al., 2018; Gliß et al., 2021). This complexity can also inhibit the interpretability of model behavior (Proske et al., 2023) and may not necessarily scale with improved model representation (Ekman, 2014). Previous studies have used idealized or simplified aerosol representations to remove this uncertainty and focus on quantifying aerosol interactions at the process level. Prescribed aerosol fields have been used to systematically quantify the sensitivity of the atmosphere to aerosol properties, including horizontal gradients (Lee et al., 2014), vertical profiles (Herbert et al., 2020; Johnson et al., 2004), concentrations (Dagan and Eytan, 2024; Tang et al., 2024), and spatial distributions (Williams et al., 2022; Dagan et al., 2021; Fiedler et al., 2017; Herbert et al., 2021a; Fiedler and Putrasahan, 2021). Idealized aerosol representations have also proven useful for identifying model structural uncertainties and estimating aerosol radiative forcing in intercomparison studies (Stier et al., 2013; Fiedler et al., 2019; Randles et al., 2013; Fiedler et al., 2023) and have been combined with reduced complexity climate models to provide a means of assessing sensitivity to future aerosol scenarios (Herbert et al., 2021b; Stjern et al., 2024; Recchia and Lucarini, 2023)."*

We have also revised a paragraph in the conclusions to positively frame our choice:

*"The idealized representation of aerosol in this model has helped identify important process-level interactions and provides a platform for future studies using realistic aerosol perturbations. The use of non-interactive aerosol may mask important feedbacks…"*

**Finally, I think the discussion around convection could be improved, but I will admit (like I do below) that this is not my area of expertise, so I cannot judge the claims sufficiently. After reading these parts a few times, I felt the arguments were wishy washy and not very convincing when it comes to convection. They strongly imply significant convection changes at times, but other times the authors caution over-interpretation. I would encourage a careful reread of those parts dealing with convection, then assessing if more careful phrasing and revision could be used. I also suggested referring (and/or reminding) the readers to a convection assessment of ICON (if at all). Would any conclusions about aerosol–convection interactions be affected if the convection itself isn't as good/robust?**

The manuscript already includes references that demonstrate the ability of our ICON configuration to represent convection. In the revised manuscript we have expanded this and refer to the relevant section later in the manuscript. See the response to comment further on.

The suppression of convection via changes to the convective environment are a consistent result throughout the analysis and are in agreement with previous studies. We only apply caution to changes in convection that are related to cloud microphysical processes (e.g., convective invigoration), which our results suggest is not consistently occurring in all regions. The role of convective invigoration remains the focus of many research groups.

To avoid confusion, we have revised the manuscript to clearly separate the responses to the convective environment from the direct modification to convection occurring within the clouds.

This includes the following revisions in Section 3.2.2:

*"Some modelling studies have suggested aerosols can also directly influence convection through invigoration of convective cloud cores via ACI, either in the liquid phase (Lebo, 2018; Sheffield et al., 2015; Fan et al., 2018) or ice phase (Heever et al., 2006; Fan et al., 2013), whilst others report suppression or regime-dependence Khain et al. (2008); Lebo and Seinfeld (2011); Storer et al. (2010); Igel and van den Heever (2021)."*

*"The latter is consistent with the impacts to the convective environment as observed over the Amazon and the Congo, while the former is a modification to the large-scale circulation."*

*"This occurs alongside an increase in IWP and θ, which suggests a role for direct modification of the convective cloud cores via convective invigoration from the cold phase (Heever et al., 2006; Fan et al., 2013)."*

And the following the conclusions section:

*"Three regions, characterized by deep convection and emissions of biomass burning aerosol, consistently demonstrated a suppression of the diurnal cycle of convection via modifications to the convective environment due to ARI and enhanced LWP due to ACI. However, the combined effect (ARI + ACI) differed in each region. The direct modification to convective clouds (suppression or invigoration) via ACI also differed between regions."*

**More comments:**

**L9: It's not readily clear to me how this sentence implies anything about atmospheric dynamics — even with controlled dynamics (say nudged simulations), we will see strong regional dependence, no? I'd recommend removing it from the abstract unless you can justify it later? I didn't really see enough strong evidence supporting it. I would remove the part about atmospheric dynamics especially that the next point is by far the least controversial and most important point of this manuscript. Note my point here isn't debating if the statement is true in general (I think it is true; e.g., I agree with your framing near L275), but rather, your manuscript/results don't have enough evidence to support "complex interplay with atmospheric dynamics" highlight imo**

Line in question: "*In our simulations over 30 days, we find that the aerosol impacts on clouds and precipitation exhibit strong regional dependence, **highlighting the complex interplay with atmospheric dynamics**"*

We agree with the reviewer. Our results suggest that most of the regional dependence seems to be driven by the large-scale environmental factors, which shape the underlying properties of the region. The reviewer refers to L275 which may be: *This demonstrates that the impact of aerosol on clouds has a diurnal driver, that may be dependent on the underlying diurnal cycle of clouds, dynamics, or solar radiation.*

As suggested, we have removed the statement.

**L10: Imho, this is the most important point of this manuscript. I think this alone justifies the manuscript and effort. It is also highlighted in the title. Great work!**

The reviewer refers to the sentence "The impact of ARI and ACI on clouds in isolation shows some consistent behaviour, but the magnitude and additive nature of the effects are regionally dependent."

We thank the reviewer for this comment.

**L13: This may benefit from a slight clarification to drive the point home stronger. I think you're trying to say something like "Because we observe pronounced diurnal cycles … we think polar-orbiting satellites may be even more limited than we already know." If so, I would rephrase to something like ", suggesting the usefulness of using polar-orbiting satellites to quantify ACI may be even more limited than presently assumed"**

Lines in question "*We also observe pronounced diurnal cycles in the response of cloud microphysical and radiative properties, suggesting a limitation of using polar-orbiting satellites to quantify or constrain aerosol-climate interactions on the diurnal scale*".

We have rewritten the sentence as suggested:

*"We also observe pronounced diurnal cycles in the response of cloud microphysical and radiative properties, which suggests the usefulness of using polar-orbiting satellites to quantify ACI and ARI may be more limited than presently assumed."*

**L17: I found the statement about ACI/ARI in the Conclusion section (e.g., L485) to be quite important and remarkable and I thus recommend including more about it in the abstract (maybe as a follow-up sentence to the great sentence on L10?)**

From the conclusions section: "*The results also strongly suggest that ACI and ARI cannot be considered independently as the responses via each pathway does not tend to be additive. Some were dominated by either ACI or ARI, and some behaved non-linearly, resulting in a response at odds with the individual components.*"

We have expanded the abstract sentence in question. The revised text is as follows:

*"The impact of ARI and ACI on clouds in isolation shows some consistent behavior, but the magnitude and additive nature of the effects are regionally dependent. Some regions are dominated by either ACI or ARI, whereas others behaved nonlinearly. This suggests that the findings of isolated case studies from regional simulations may not be globally representative and that ARI and ACI cannot be considered independently and should both be interactively represented in modelling studies."*

**L86: Usually, models prescribe the sea ice extent, but let the sea-ice thermodynamics run. If that's the case in your model, I would simply add "extent" or "cover" after sea ice to avoid confusion**

The line in question is "We run the model in an atmosphere-only mode, with oceanic properties (sea surface temperature and sea ice) prescribed following the atmospheric model intercomparison project AMIP"

The model does not have a sea-ice thermodynamic model and instead uses the prescribed sea-surface properties as boundary conditions, in-line with the AMIP protocol. We have rephrased this sentence as follows:

*"We run the model in an atmosphere-only mode, with sea surface properties (sea surface temperature and sea ice **concentration**) prescribed **as atmospheric boundary conditions** following the atmospheric model intercomparison project AMIP"*

**L117: I would mention the exact process (which you do later anyway, but might as well do it here too: autoconversion)**

Line in question: "*In this study, aerosols are represented using the simple plume implementation of theMax Planck Institute Aerosol Climatology version 2 (Stevens et al., 2017, MACv2-SP), which is used in ICON to represent aerosols in the radiation scheme. We extend this to the cloud microphysics scheme to link changes in aerosol to the warm-rain process*".

This has been included as suggested.

**L118: I thought they are non-interactive but spatially and temporally varying? Like AOD is different over Africa compared to Poles. Same for Nc and/or Nd. Are you talking about something else here? When I read "fields … provided by MACv2-SP" I think of fields that MACv2-SP prescribes for your simulations (optical depths + Nd + Nc). Maybe you're talking about the inputs to the plume model or something else here…**

Line in question: "*The prescribed fields of aerosol provided by MACv2-SP are non-interactive and spatially invariable, but magnitudes are temporally variable.*"

The reviewer is correct: the aerosol fields provided by MACv2-SP are spatially and temporally variable. The phrasing above was aimed to clarify that once prescribed, the spatial distribution is set. This is misleading, so has been rewritten for clarity:

*"The prescribed fields of aerosol provided by MACv2-SP are non-interactive, but magnitudes are spatially and temporally variable."*

**L164: Minor point: is it more or less consistent with obs than the prior tuning?**

This refers to the parameters $a_N$ and $b_N$ which are used to calculate $N_{d,cld}$ from the AOD. In response to the second reviewer, we expanded the $N_d$ climatology in Figure 1b. In the

Grosvenor et al. (2018) manuscript, $N_d$ was shown over oceans only, but the authors also provide data over land in the supporting information. This provides a better test for our aerosol representation as the MACv2-SP plumes are centered over land (where primary aerosol emissions occur), not the ocean. We used this alternative dataset and removed grid boxes with relatively large uncertainty - where the number of days with a successful retrieval over the 13 year time series was less than 50.

[Figure]

Figure R1. Revised annual mean $N_d$ climatology from Grosvenor et al. (2018). Blank regions show grid boxes where there were fewer than 50 retrievals in the 13 year time series.

A statistical comparison to the climatology using the new and default parameters is shown in Table R1 below. All statistical tests demonstrate that the new parameters are more appropriate than the default ones. Using the more expansive $N_d$ climatology has also increased the correlation coefficient from 0.34 (oceans only - unconstrained) to 0.57 (all available grid points - constrained to > 50 data points). This value still demonstrates some discrepancy between simulation and observations but provides more confidence to the reader.

|  | Herbert et al. (2021) parameters | Default (Stevens et al., 2017) parameters |
|---|---|---|
| Root mean square error | 49.5 | 70.6 |
| Mean absolute error | 34.0 | 47.3 |
| Normalized mean bias | -0.12 | -0.35 |
| Correlation coefficient | 0.57 | 0.42 |

Table R1. Statistical tests between observed $N_d$ from Grosvenor et al. (2018) and simulated $N_d$ using the MACv2-SP plume model.

The revised climatology has been included in the revised manuscript, and the following text has been included in Sect. 2.3:

*"A comparison between simulated and observed $N_d$ yields a root mean square error (RMSE) of 49 $cm^{-3}$ and a correlation coefficient of 0.57 (the default parameters $a_N$ and $b_N$ yield an RMSE of 70 $cm^{-3}$ and correlation coefficient of 0.42). The discrepancy is in part due to high simulated values over biomass burning regions that are not reflected in annual mean observations, but also due to regional variability that MACv2-SP does not capture (e.g., North America)."*

Figure 1b has also been updated to Fig R1 above and the caption has been expanded to describe the new method:

*"... and (b) annual mean $N_d$ for cloud tops < 3.2 km (2003 – 2015; only showing grid points with 50 successful retrievals) estimated by Grosvenor et al. (2018)..."*

**L172: Minor: I would consider using the table to specify the details very clearly and anchor the discussion around it. For example, PI run uses Nd,cld that's constant (value xyz), but Nd,rad that evolves according to Eq X with constant values for xx and yy. Currently, you do have all the info in the text, it is just slightly hard to parse. But feel free to ignore…**

This is an excellent suggestion. Table 1 has been expanded and provides more detail. We have also rewritten the opening paragraph of Section 2.3:

*"We use four simulations to explore the role of aerosols on clouds and climate (outlined in Table 1). The control simulation (PI) uses values that are representative of a pre-industrial atmosphere consisting of natural aerosol and background ARI and ACI effects. Global fields of natural aerosol extinction are represented by the K19 climatology for the year 1850. $N_{d,cld}$ is held constant at a value of 80 $cm^{-3}$, whilst $N_{d,rad}$ follows a vertical profile according to Eq. 1 and varies spatially with $N_{d,rad-sfc}$ set to 120 $cm^{-3}$ on land and 80 $cm^{-3}$ over oceans. A second simulation (PD) is run with values that are representative of a present-day atmosphere that includes ACI and ARI effects due to anthropogenic activity. Aerosol extinction fields from anthropogenic aerosol are represented by the plume model MACv2-SP for the year 2016 and added to the pre-industrial contribution. The spatial distributions of $N_{d,cld}$ and $N_{d,rad}$ are modified using the scaling factor $f_N$ (Eq. 2), which varies spatially with the anthropogenic aerosol contribution. The third and fourth simulations are used to isolate ACI and ARI effects in the present-day atmosphere. In the third simulation, $PD_{ARI}$, extinction from the anthropogenic aerosols are included, but the scaling factor $f_N$ is not applied to $N_{d,cld}$ and $N_{d,rad}$; this isolates ARI effects associated with anthropogenic aerosol. In the final simulation, $PD_{ACI}$, the ACI scaling factor $f_N$ is applied, but aerosol extinction remains at pre-industrial values; this isolates ACI effects associated with anthropogenic aerosol."*

**L200–215: Great! I think this is very promising and appropriate!!**

This comment refers to the lines where we outline that we do not focus on global aerosol forcing estimates, and instead "*exploit the capability of the model to represent scales that are traditionally used by high-resolution simulations*".

This was the new focus of the study as suggested by all three reviewers. We are very happy that the reviewer positively acknowledges these revisions.

**L301 (and many other places): You use the word "response" a lot in this manuscript (on one editor, I found 117 instances when I used the pdf search functionality) and you use it to mean at least two different things. On this line, you likely mean just the values of the CFtotal in Figure 8. But elsewhere you use it to mean the PD-PI response (other places you also use in the second sense, related to change). Can you try to clarify this throughout the manuscript? For example, in Figure 7 (just above) you likely just mean "… during diurnal cycle of maximum absolute for each..." And on this line you like just mean "spatially consistent with CFtotal, with" … unless I missed something obvious?! I bring this up because the word response tripped me up multiple times, and I got confused trying to understand what you're trying to say. I found myself having to go back and cross-check. My recommendation is to use the word "response" only in making an explicit point about "a response to a perturbation" (e.g., your L308, or caption of Figure 10). For everything else, I'd use different ways to describe the signal more matter-of-factly/plainly**

We use the word response to refer to the change in the variable due to the aerosol perturbation (e.g. PD - PI). In the three examples used in the reviewer comment we are indeed referring to the response. In response to this comment, we have gone through the manuscript to break up the repetitive use of the word response. In some instances response has been changed to the 'aerosol effect' and in others the 'change in X' or 'ΔX'. We have also checked for instances where its use is either redundant or potentially misleading - for example, we often use the term "...in the PD response". These have been revised for clarity.

**L319: Minor, but has there been an assessment of deep convection characteristics in this model? Is it okay compared to obs? I am not a convection expert and I don't study convection–aerosol interactions, but I keep hearing that 5km models have pretty severe convection problems, so I am not sure how informative it is to study the aerosol impacts on or due to convection, if the convection is poorly simulated to begin with. This is borderline outside your scope here, so my only minor request is to add (somewhere in the manuscript, if not added already) some references about a convection assessment of this specific model config if one exists…**

Line in question: The novel aspect of this study is the globally resolved deep convection, hence we primarily focus on regions with deep convection.

Following comments in the first round of reviews, we expanded a paragraph in the introduction to discuss the model's ability to represent convection. The paragraph is as follows:

*"This configuration of ICON does not explicitly resolve the smallest scales of convection (< 5 km) but has been shown to reproduce observed diurnal and seasonal cycles of tropical precipitation (Segura et al., 2022). Given that ESMs tend to use spatial resolutions of tens to hundreds of kilometres, this makes a marked improvement in our ability to resolve many aspects of convection (Done et al., 2004; Prein et al., 2013) and is well suited for our study."*

We have slightly expanded this as follows and included a pointer to the section (Sect. 2.1) on the relevant line.

"*This configuration of ICON does not explicitly resolve the smallest scales of convection (< 5 km) but has been shown to reproduce **many features of the climate system relevant for this study (Hohenegger et al., 2023), including seasonal cycles of precipitation and soil moisture, the structure of the atmosphere in deep convective regions, and coupling between sea surface temperature and precipitation.** Segura et al., (2022) also demonstrate that this configuration reproduces the observed diurnal cycle of tropical precipitation. Given that ESMs tend to use spatial resolutions of tens to hundreds of kilometres, this makes a marked improvement in our ability to resolve many aspects of convection (Done et al., 2004; Prein et al., 2013) and is well suited for our study.*"

**L375 (and thereabouts): I don't think these variables (Mflux, W\*) were defined/introduced, but I may have missed them… I would introduce them with equations or a reference to an equation/methodology elsewhere**

$M_{flux}$ is defined in Section 3.1 but W\* is only defined in a figure caption. The latter has been moved to the main text along with an introduction to the other variables shown in Figures 11, 12, and 13:

*"In Figures 11 – 13 we focus on the drivers of the cloud response to anthropogenic aerosol in the three convective regions. Variables include IWP, P, and $M_{flux}$, and profiles of ice water content (IWC), liquid water content (LWC), potential temperature (θ), water vapor (Qv) and vertical velocity (W\*) calculated in regions characterized by ascent (1° grid boxes where the mean vertical velocity at 300 hPa during the PI simulation is positive). The frequency of output on all vertical…"*

**L397: which state? Here and elsewhere, I would try to be explicit about which state you mean (dynamic state, thermodynamic state, aerosol state, cloud state, or general atmospheric state, or something totally different?)**

Line in question: "*The Congo and Amazon regions respond consistently to ARI and ACI individually, but the total responses are different, suggesting a degree of state dependence*".

These instances have been clarified as *thermodynamic states* and *thermodynamic state dependence*.

**L415: Related to the above: you start with localized modification to the convective environment, but here you say convection itself is modified. How did the jump take place? Is the reader supposed to assume that the localized modifications to the convective environment will always lead to convection changes?**

**L410: Like above, here and elsewhere, I would try to be explicit about what you mean by convective environment. Of course, you can define it the first time you mention it and say "convective environment" is my shorthand for what I just defined throughout the manuscript.**

We have combined these two comments.

The *convective environment* refers to the properties of the local atmosphere that describe the potential for initiation and development of convection. This will include vertical profiles of temperature, moisture, and wind shear, and larger-scale horizontal fluxes (convergence/divergence). If this environment is made more or less favourable for the initiation and/or development of convection then it follows that there will be changes to convection. The aerosol perturbations can modify these properties through changes to surface fluxes and heating profiles. The aerosol may also directly modify convection via cloud microphysical processes that are linked to the availability of aerosol (CCN, cloud droplets, latent heat, buoyancy).

As suggested, we have defined the term *'convective environment'* upon first use:

*"However, this will also be sensitive to the* *thermodynamic properties of the region that provide the potential for convection (the convective environment),* *the different aerosol plume characteristics, or buffering of the response due to coupling to large-scale meteorology."*

We have also expanded/revised some sentences for clarity and to help distinguish the convective environment from convection itself:

*"The strongly absorbing aerosol produces localized heating of the smoke layer, suppressing mixing in the lower atmosphere and drying aloft,* *which reduces the potential for convection in the region.* *The suppressed convection reduces the regional-mean vertical extent of clouds and decreases LWC throughout the column."*

*"...suggests the differences may be due to stronger capacity to buffer the perturbation over the Congo, which tends to exhibit more convection than the Amazon, or differences in the convective environments Storer et al. (2010)* *that may result in one region being more susceptible to the aerosol perturbation.**"*

*"The contrasting roles of ACI and ARI in the Congo and the Amazon* *suggest that the response of convection to changes in the aerosol population is dependent on the background thermodynamic state and convective environment,* *which has also been observed in remote-sensing studies of the Amazon region.."*

*"The primary driver is a localized modification to the convective environment that* *suppresses convection and* *reduces daily accumulated P…"*

**L498: I think I finally get what you're saying with the "spatially invariable" part… the spatially invariable aerosol as input to MACv2-SP? But this is quite misleading! Your model doesn't really care about that aerosol input to the MACv2-SP model; what your model cares about is the effect of the aerosol (as proxied by radiation properties like**

**optical depths, Nd,cld, and Nd,rad). These are definitely spatially variable! Am I misunderstanding something here?**

> This comment was echoed previously. The aerosol fields provided by MACv2-SP are spatially and temporally variable. The phrasing of 'spatially invariable' was aimed to clarify that once prescribed, the spatial distribution is set. This is misleading, so has been rewritten for clarity.

**L502: I would delete this last sentence advertising modeling groups (doesn't really add any context beyond what you said)**

> As suggested, the sentences have been removed.

**L520: I would remove this entire paragraph (ending on the prior paragraph is much better imho).**

> As suggested, the paragraph has been removed.

**L527: Could you provide a specific commit/sha for the code you used? Or even better include it in a permanent Zenodo repository with its own DOI? Thanks!**

> This information has been added to the data availability section.
* * *
**Reviewer 2**

**The authors addressed some of my comments in the last round of review, but the manuscript still suffers from severe flaws despite being mostly rewritten.**

**Firstly, the authors used LOESS, a tool for seasonal-trend decomposition, to separate the long-term component from the short-term component in the model time series. They insisted that the long-term component comprised internal variability and persistent responses (Lines 225-227), while the short-term component captured the diurnal responses. I disagree that LOESS can separate model internal variability from aerosol impacts. LOESS is just a statistical tool. If it could filter out model internal variability, scientists wouldn't need to run hundreds of years of simulations or conduct multiple ensemble simulations to minimize model uncertainties. Aerosol signals can accumulate and interact with the atmosphere. While I understand the limitations of 1-month simulations in this study due to computational resources, the authors didn't exploit the existing model results for deep analyses. Most of the analyses are superficial and lack depth and evidence. Instead of relying on these statistical analyses, I suggest the authors delve into model physics and provide a process-level analysis of how global convection-**

**permitting simulations change or validate our understanding of aerosol-cloud and aerosol-radiation interactions.**

We thank the reviewer for this comment but disagree with all of the points raised. We believe our method and analysis are wholly appropriate, robust, and provide original evidential insight and conclusions that are supported by previous studies.

We do not aim to reproduce and separate the full natural variability associated with our perturbation as this is impossible with our fixed SST configuration and relatively short timescales. However, this is not the aim of our study nor the timescale we are focusing on. Instead, our study implicitly focuses on the 'rapid adjustments' of convectively active regions to aerosol perturbations, which is a very different question from studies that aim to capture the full climate response to anthropogenic aerosols. For example, Shipeng et al. (2021) ran 100 year simulations with a mixed-layer slab ocean configuration in order to quantify both 'fast' and 'slow' responses to idealized aerosol perturbations.

From previous studies we know that these rapid adjustments occur within the diurnal cycle, so we expect our response to contain a 'repeatable' signal on a diurnal time scale. Most time series decomposition methods are rigid and not designed for a signal that varies day to day - so inappropriate for a background meteorological state that will vary day to day. The LOESS method allows for these variations and provides an excellent tool for this study.

> From Dokumentov and Hyndman (2022): *"Existing time series decomposition methods are designed for monthly and quarterly data with few tools available for more frequent data. The seasonal-trend decomposition using Loess (STL) procedure of Cleveland et al. (1990) is the only widely available decomposition tool for data observed more frequently than monthly."*

The short-term component from LOESS provides the diurnally varying response throughout the time series. This fluctuates around a value of zero so misses any component of the response that is persistent. Our method attempts to extract this from the LOESS long-term component that will implicitly contain any model response that is not purely due to the aerosol perturbation as we discuss in the manuscript (e.g internal variability). We do not claim this is a perfect extraction and state this in the manuscript (we make this clearer in the revised manuscript - see below).

Though the technique is often referred to differently, we are not the first to use LOESS to decompose responses in the climate system. The power of this statistical decomposition tool (and other similar methods) has been well demonstrated by other studies focusing on the climate, e.g., Deng and Fu, 2019; Carslaw, 2005; Verbesselt et al., 2010; He et al., 2022; Liu and Zhang, 2024; Zhou et al., 2015; Cleveland, 1979; da Silveira Bueno et al., 2024; Papacharalampous et al., 2018; Quan et al., 2016; Jaber et al., 2020; Rabbi and Kovács, 2024; Moradi, 2022; Deng et al., 2015.

We have revised the manuscript to address the reviewer's concerns. First, throughout the manuscript (abstract, introduction, conclusions) we now clearly state that we are focusing on the rapid response due to the aerosol perturbation. Second, we have revised Sect 2.4 (Temporal decomposition of regional response) to clarify that the isolation of internal variability is approximate. We also include more description of the LOESS method, its

applicability to our research question, and include citations to several previous studies. Finally, we acknowledge the limitations associated with the temporal decomposition method in the conclusions.

In Sect. 2.4:

We attempt to isolate the responses due to the aerosol perturbation from internal variability and noise by temporally decomposing…

*"LOESS is a statistical decomposition tool that can be applied to extract responses occurring on relatively high frequencies (e.g. diurnal) and has been used in previous climate-focused studies (e.g. [citations…])."*

*"...may represent an important aerosol effect, hence we* attempt *to recapture this using a second application of the decomposition tool"*

*"This method assumes that any internal variability is evenly distributed around the time-independent response, which may not be true, but provides a reasonable approximation* and should capture regions where strong persistent responses occur."

In the conclusions:

*"In an effort to isolate the aerosol impacts from internal variability, we subset the globe"*

*"Future studies should also consider building on the temporal decomposition method (Sect. 2.4) as not all internal variability can be isolated from the aerosol-driven response. The method assumes that mean internal variability during the time series is equal to zero; whilst this may be true on very long time scales (years to decades) it is unlikely to be the case over our simulation duration. The method additionally assumes that the persistent response due to the aerosol perturbation is independent of time. In reality, this component may increase or decrease during the simulation due to local or non-local feedbacks between clouds, the surface, and the thermodynamic properties of the region. This could be explored in future studies with longer simulations."*

**Secondly, the authors divided the globe into 288 15°×15° regions and treated the global simulation as a sum of 288 regional simulations. While this approach is acceptable, it has limitations. The authors should discuss the limitations, especially the interactions between nearby 15°×15° regions. The transport effect should be significant in the 30-day simulation.**

We did not perform 288 separate simulations, rather we took the outputs from the global simulations and subset them into 288 defined regions. Using our method, the mean global response will be the same whether it has been calculated from variables on the native grid (~5 km) or from the 15°×15° grid. The regional subsetting is performed in order to encompass a large enough number of native data points to maximise the use of the LOESS temporal decomposition method.

Unlike traditional regional-scale simulations, our configuration implicitly allows neighbouring regions to interact with each other. Thus, the transport of energy, moisture, etc to neighbouring regions (and further afield) can be seen to be included as boundary

conditions. The cloud response due to the aerosol perturbation in one region will indirectly influence the response in the neighbouring region, and so on. We view this as a benefit as it permits non-local regional responses to an aerosol perturbation, aiding in the realism of the response.

In the revised manuscript we have clarified how the '288 equivalent simulations' were achieved by taking the global outputs and then subsetting into regions:

*"To study aerosol impacts on the global scale we subset the outputs from the global simulations into 15°×15° regions, producing the equivalent of 288 regional-scale simulations running for a 30 d period. With this method, the regions can interact with each other, and any regional aerosol response is transported to neighboring regions."*

*"In a global-scale analysis we subset the global simulation outputs into 15°×15° regions, producing the equivalent of 288 regional high-resolution simulations that can interact with each other."*

**Thirdly, as mentioned in the first major comment, the analyses lack depth and robust evidence. This issue was raised in the first round of review but was not significantly improved in the revised manuscript. The four 1-month simulations are complete, and I suggest the authors think carefully about how to exploit the existing simulations for a comprehensive analysis.**

Our analysis addresses our objectives and provides an original perspective into the variable role of aerosol in the climate system. The global-scale analysis is consistent with literature and the process-level analysis supports our conclusions, which are wholly in-line with literature from the community. We acknowledge in the manuscript that there are some limitations to the study design, but these are unlikely to greatly affect our core conclusions. This study has provided an excellent starting point for us to include interactive aerosols in the ICON model and other kilometre-scale modelling configurations and will help shape the design of future studies and model intercomparison projects.

**Fourthly, some sentences are difficult to understand. I had to read the manuscript several times to grasp (or guess) the authors' intentions. If the authors plan to revise the manuscript, language improvement is necessary.**

Following this, and a similar suggestion from Reviewer 1, we have revised the manuscript to improve the language and readability.

**Since most of the analyses are based on the LOESS decomposition, I won't comment further on those results, even though I agree with some conclusions. Please find below additional detailed comments.**

**Title: How about "Regional variability of aerosol impacts on …"? "Regional-scale aerosol impacts" is misleading.**

We agree with the reviewer and have revised the title as suggested.

**Lines 31-43: Please consider reorganizing these sentences.**

As suggested by the reviewer, the third, fourth and fifth paragraphs in the introduction (which include lines 31-43) have been reordered and revised to provide a more logical story.

**Lines 65-67: Your simulations seem not to be able to reproduce the observations, as shown in Figure 4. Is my understanding correct?**

The sentence in question is "[Sato et al. 2018] found that using an explicit representation of cloud microphysics on a global scale produced a negative LWP-AOD relationship, in agreement with satellite observations, that was not replicated in a coarser global model."

We use this study in the introduction as an example of how model resolution can impact the representation of cloud microphysical processes, rather than to characterise the relationship between LWP and Nd. The annual-mean relationship between LWP and Nd in liquid topped clouds (as shown in Sato et al. (2018)) is known to be strongly nonlinear (Gryspeerdt et al., 2019) and difficult to quantify from remote satellite observations (Jia et al., 2022; Grosvenor et al., 2018; Quaas et al., 2020). Gryspeerdt et al. (2019) and others have found that at low values of Nd the dLWP/dNd relationship is likely positive, and at higher values of Nd the relationship is likely negative. A linear trend between extreme ends of the Nd distribution hides this complexity and does not consistently result in a negative trend as observed by Sato et al. (2018). The magnitude and sign of the trend is sensitive to the satellite product (Gryspeerdt et al., 2019; Grosvenor et al., 2018), the region of interest (Michibata et al., 2016), and meteorological variables (Michibata et al., 2016; Sato et al., 2018). Additionally, as we conclude in our manuscript, remote sensing observations may misrepresent the daily-mean response of clouds to aerosol perturbations as they only observe a short time window in the diurnal cycle, as demonstrated by Figure 10.

Therefore, the observations of the LWP-Nd relationship presented by Sato et al. (2018), and their comparison method, are not directly applicable to our simulation but do provide an interesting point of discussion. The most relevant clouds to compare are those in the marine stratocumulus region of the SE Atlantic. Although we observe a diurnal cycle in the LWP response to the aerosol here (Figure 4c, Figure 10f) it is always positive when the persistent effect is included (Figure 6b and 6c). This is consistent with Michibata et al. (2016) but inconsistent with Sato et al. (2018).

Gryspeerdt et al. (2019) suggest that the contrasting LWP-Nd relationship at low vs high Nd is related to the switch from precipitating to non-precipitating clouds. Sato et al. (2018) also show they were able to obtain a negative LWP-Nd relationship when the representation of precipitation was improved. This suggests our LWP response may be sensitive to the representation of precipitation and its link to Nd, which is simplified in our model (autoconversion).

We include the following in Section 3.2.1:

"The positive relationship between ΔLWP and $\Delta N_{d,cld}$ in the SE Atlantic is consistent with remote sensing observations from Michibata et al. (2016), but inconsistent with those from Sato et al. (2018), and may be sensitive to the representation of the warm-rain process (Gryspeerdt et al., 2019; Sato et al., 2018, Terai et al., 2020); we revisit this in the conclusions."

And include the following in the Conclusions section:

"Additional sources of uncertainty arise from the cloud microphysics scheme and unresolved convection. The choice of cloud microphysics scheme and representation of cold-phase processes have been shown to impact the sensitivity of convective clouds to aerosol (Heikenfeld et al., 2019; White et al., 2017; Sullivan and Voigt, 2021; Marinescu et al., 2021), while the representation of the warm-rain process and its link to aerosols have been shown to be important for ACI impacts on warm-phase clouds (Gryspeerdt et al., 2019; Sato et al., 2018, Terai et al., 2020)."

**Line 85: What is JSBACH?**

This acronym has now been defined as follows:

".. the Jena Scheme for Biosphere Atmosphere Coupling in Hamburg (JSBACH) dynamic vegetation model"

**Lines 107-109: I am still confused about your definition of biomass burning. Do you consider it an anthropogenic source? If not, the selection of September due to its remarkable biomass burning does not make sense since your study focused on the impact of anthropogenic aerosols. It doesn't hurt your experimental design, but please describe the model setup and reasoning as accurately as possible.**

**Lines 128-131: Again, it is confusing whether biomass burning is considered natural or anthropogenic sources in your study.**

**Lines 135: What do you mean by the natural and anthropogenic contributions of the biomass burning plumes?**

**Figure 1 and Lines 187-188: It seems that biomass burning is considered anthropogenic sources in this study. I must disagree with that. It would be better to clarify this point and use more accurate wordings throughout the manuscript to distinguish the PD and PI experiments.**

In our simulation we assume that anthropogenic activity contributes to emissions of absorbing aerosol, which are represented in MACv2-SP. This is consistent with global databases (van der Werf et al., 2017), CMIP5 and CMIP6 inventories (Lamarque et al., 2010; van Marle et al., 2017), and observations (e.g., Abatzoglou and Williams, 2016; Knorr et al., 2016). The magnitude is uncertain (e.g., Hamilton et al., 2018; Lauk and Erb, 2009) and we use a value at the higher end of this uncertainty in order to maximise the signal.

In this study, we are focused on understanding the regional variability of the role that idealized aerosol perturbations have on clouds and climate, which we frame within the context of anthropogenic activity. We have revised the manuscript to make this point clearer. The aim is to make the reader aware that we are focused on the response of clouds and climate to an idealized aerosol perturbation, for which we use a present-day vs pre-industrial comparison. We have included the following sentence at the end of the introduction:

*"We analyze the response of clouds and the thermodynamic environment to an aerosol perturbation by contrasting simulations using aerosol representative of the pre-industrial era with aerosol representative of the present-day."*

In the rest of the manuscript, we refer to the change in aerosol as an aerosol perturbation (as opposed to anthropogenic aerosol).

**Lines 117-118: What do you mean by non-interactive and spatially invariable but temporally variable? In Line 95, you mentioned that Nd follows a predefined vertical profile. If aerosol concentrations change, will Nd change? How about the vertical profile shape of Nd?**

This section has been revised for clarity.

**Lines 125-128: Do you mean the 3D fields of aerosol extinction are calculated based on the nine plumes of aerosol concentrations and optical properties?**

The lines mentioned are: *"MACv2-SP, described in full by Stevens et al. (2017), provides the model with 3D fields of aerosol extinction that are predefined at the beginning of the simulation. The fields are represented as nine plumes spatially consistent with the dominant sources of global anthropogenic aerosol emissions. Each plume is characterized by parameters that control its horizontal and vertical distribution, aerosol concentration and optical properties, annual cycle, and year-to-year variations."*

For clarity, we have rewritten this as suggested by the reviewer:

*"Anthropogenic aerosol perturbations are represented using MACv2-SP, described in full by Stevens et al. (2017), which provides the model with 3D fields of aerosol extinction that are calculated for nine predefined plumes of aerosol concentrations and optical properties. The plumes are spatially consistent with the dominant sources of global anthropogenic aerosol emissions, and each is characterized by parameters that control its horizontal and vertical distribution, aerosol concentration and optical properties, annual cycle, and year-to-year variations."*

**Line 140: Please explain how the change in the ratio of natural to anthropogenic aerosols can provide a stronger signal if you only change the anthropogenic aerosol concentrations from PI to PD? The current model description is confusing, and I don't understand the logic.**

We have revised the manuscript to provide clearer information on how the two sources of aerosol (natural and anthropogenic) are represented, and how we enhance the anthropogenic contribution from the biomass burning plumes in MACv2-SP.

**Line 163-164: No. Figure 1 shows the overestimation of Nd compared to the annual climatology.**

We have revised our observational climatology to ensure we are making a like-for-like comparison and have expanded the $N_d$ climatology in Figure 1b. In the Grosvenor et al. (2018) manuscript, $N_d$ was shown over oceans only, but the authors also provide data over land in the supporting information. This provides a better test for our aerosol representation as the MACv2-SP plumes are centered over land (where primary aerosol emissions occur), not the ocean. We used this alternative dataset and removed grid boxes with relatively large uncertainty - where the number of days with a successful retrieval over the 13 year time series was less than 50. The revised climatology is shown below in Fig. R1 (duplicated from comment from Reviewer 1).

[Figure]

Figure R1. Revised annual mean $N_d$ climatology from Grosvenor et al. (2018). Blank regions show grid boxes where there were fewer than 50 retrievals in the 13 year time series.

A statistical comparison to the climatology using the new and default parameters is shown in Table R1 below (also duplicated from previous comment). Using the more expansive $N_d$ climatology has increased the correlation coefficient from 0.34 (oceans only - unconstrained) to 0.57 (all available grid points - constrained to > 50 data points). This value still demonstrates some discrepancy between simulation and observations but provides more confidence to the reader. We also stress, as we do in the manuscript, that this is representative of the spatial distribution and magnitude of anthropogenic aerosol, and appropriate for our study.

|  | Herbert et al. (2021) parameters | Default (Stevens et al., 2017) parameters |
|---|---|---|
| Root mean square error | 49.5 | 70.6 |
| Mean absolute error | 34.0 | 47.3 |
| Normalized mean bias | -0.12 | -0.35 |
| Correlation coefficient | 0.57 | 0.42 |

Table R1. Statistical tests between observed $N_d$ from Grosvenor et al. (2018) and simulated $N_d$ using the MACv2-SP plume model.

The revised climatology has been included in the revised manuscript, and the following text has been included in Sect. 2.3:

*"A comparison between simulated and observed $N_d$ yields a root mean square error (RMSE) of 49 cm$^{-3}$ and a correlation coefficient of 0.57 (the default parameters $a_N$ and $b_N$ yield an RMSE of 70 cm$^{-3}$ and correlation coefficient of 0.42). The discrepancy is in part due to high simulated values over biomass burning regions that are not reflected in annual mean observations, but also due to regional variability that MACv2-SP does not capture (e.g., North America)."*

Figure 1b has also been updated to Fig R1 above and the caption has been expanded to describe the new method:

*"... and (b) annual mean $N_d$ for cloud tops < 3.2 km (2003 – 2015; only showing grid points with 50 successful retrievals) estimated by Grosvenor et al. (2018)..."*

We give reasons for their discrepancy and provide reasoning as to why this isn't a primary concern in the study.

**Line 168: I guess Nd,cld is three-dimensional. So you enhance Nd,cld everywhere throughout the vertical column with the same factor. Is my understanding correct?**

Yes this is correct. We have rewritten the paragraph to make this clear. We have also emphasized that this produces an idealized distribution of Nd.

*"In the default ICON setup, the microphysics scheme uses a predefined value for the cloud droplet number concentration ($N_{d,cld}$) that is spatially invariable and constant in altitude. We use this for our PI distribution of $N_{d,cld}$, which we set to 80 cm$^{-3}$. We represent ACI effects in the microphysics scheme using the ACI scaling factor $f_N$ from MACv2-SP, as calculated above. Applying $f_N$ to the pre-industrial distribution of $N_{d,cld}$ provides an idealized present-day distribution that is spatially consistent with the anthropogenic contributions in the MACv2-SP plumes."*

We also highlight this limitation in the conclusions:

*"The idealized representation of aerosol and N$_d$ in this model has helped identify important process-level interactions and provides a platform for future studies using realistic aerosol perturbations. The use of non-interactive aerosol may mask important feedbacks and processes including the impact of clouds and precipitation on the spatio-temporal distribution of aerosols, changes to the surface properties and energy fluxes, and turbulence that would influence emissions and aerosol removal processes. Changes in aerosol concentrations would also affect N$_d$ concentrations and vertical profiles."*

**Lines 179-182: Figure 1 can't distinguish anthropogenic from natural aerosols. It would be better to add a supplemental figure showing the anthropogenic and natural aerosol emissions to support this statement.**

This has now been included in the supporting information (Sect. S1 and Fig. S1) and referenced in Sect. 2.3.

**Additional references**

Abatzoglou, J. T. and Williams, A. P.: Impact of anthropogenic climate change on wildfire across western US forests, Proceedings of the National Academy of Sciences, https://doi.org/10.1073/pnas.1607171113, 2016.

Carslaw, D. C.: On the changing seasonal cycles and trends of ozone at Mace Head, Ireland, Atmospheric Chemistry and Physics, https://doi.org/10.5194/acp-5-3441-2005, 2005.

Cleveland, W. S.: Robust Locally Weighted Regression and Smoothing Scatterplots, Journal of the American Statistical Association, https://doi.org/10.1080/01621459.1979.10481038, 1979.

da Silveira Bueno and co-authors: Global warming and coastal protected areas: A study on phytoplankton abundance and sea surface temperature in different regions of the Brazilian South Atlantic Coastal Ocean, Ecology and Evolution, https://doi.org/10.1002/ece3.11724, 2024.

Dagan, G. and Eytan, E.: The Potential of Absorbing Aerosols to Enhance Extreme Precipitation, Geophysical Research Letters, https://doi.org/https://doi.org/10.1029/2024GL108385, 2024.

Dagan, G. and co-authors: An Energetic View on the Geographical Dependence of the Fast Aerosol Radiative Effects on Precipitation, Journal of Geophysical Research: Atmospheres, https://doi.org/https://doi.org/10.1029/2020JD033045, 2021.

Deng, J. and co-authors: Long-term changes in surface solar radiation and their effects on air temperature in the Shanghai region, International Journal of Climatology, https://doi.org/https://doi.org/10.1002/joc.4212, 2015.

Deng, Q. and Fu, Z.: Comparison of methods for extracting annual cycle with changing amplitude in climate series, Climate Dynamics, 5059–5070, https://doi.org/10.1007/s00382-018-4432-8, 2019.

Dokumentov, A. and Hyndman, R. J.: STR: Seasonal-Trend Decomposition Using Regression, INFORMS Journal on Data Science, https://doi.org/10.1287/ijds.2021.0004, 2022.

Ekman, A. M. L.: Do sophisticated parameterizations of aerosol-cloud interactions in CMIP5 models improve the representation of recent observed temperature trends?, Journal of Geophysical Research: Atmospheres, https://doi.org/https://doi.org/10.1002/2013JD020511, 2014.

Fiedler, S. and Putrasahan, D.: How Does the North Atlantic SST Pattern Respond to Anthropogenic Aerosols in the 1970s and 2000s?, Geophysical Research Letters, https://doi.org/https://doi.org/10.1029/2020GL092142, 2021.

Fiedler, S. and co-authors: Historical Changes and Reasons for Model Differences in Anthropogenic Aerosol Forcing in CMIP6, Geophysical Research Letters, https://doi.org/https://doi.org/10.1029/2023GL104848, 2023.

Gryspeerdt, E. and co-authors: Constraining the aerosol influence on cloud liquid water path, Atmospheric Chemistry and Physics, https://doi.org/10.5194/acp-19-5331-2019, 2019.

He, R. and co-authors: Modeling and predicting rainfall time series using seasonal-trend decomposition and machine learning, Knowledge-Based Systems, https://doi.org/10.1016/j.knosys.2022.109125, 2022.

Herbert, R. and co-authors: Nonlinear response of Asian summer monsoon precipitation to emission reductions in South and East Asia, Environmental Research Letters, https://doi.org/10.1088/1748-9326/ac3b19, 2021.

Jaber, S. M., , and Abu-Allaban, M. M.: MODIS-based land surface temperature for climate variability and change research: the tale of a typical semi-arid to arid environment, European Journal of Remote Sensing, https://doi.org/10.1080/22797254.2020.1735264, 2020.

Jia, H. and co-authors: Addressing the difficulties in quantifying droplet number response to aerosol from satellite observations, Atmospheric Chemistry and Physics, https://doi.org/10.5194/acp-22-7353-2022, 2022.

Johnson, B. T. and co-authors: The semi-direct aerosol effect: Impact of absorbing aerosols on marine stratocumulus, Quarterly Journal of the Royal Meteorological Society, https://doi.org/https://doi.org/10.1256/qj.03.61, 2004.

Knorr, W. and co-authors: Air quality impacts of European wildfire emissions in a changing climate, Atmospheric Chemistry and Physics, https://doi.org/10.5194/acp-16-5685-2016, 2016.

Lamarque, J.-F. and co-authors: Historical (1850–2000) gridded anthropogenic and biomass burning emissions of reactive gases and aerosols: methodology and application, Atmospheric Chemistry and Physics, https://doi.org/10.5194/acp-10-7017-2010, 2010.

Lee, S. S. and co-authors: Effect of gradients in biomass burning aerosol on shallow cumulus convective circulations, Journal of Geophysical Research: Atmospheres, https://doi.org/https://doi.org/10.1002/2014JD021819, 2014.

Liu, X. and Zhang, Q.: Combining Seasonal and Trend Decomposition Using LOESS With a Gated Recurrent Unit for Climate Time Series Forecasting, IEEE Access, https://doi.org/10.1109/ACCESS.2024.3415349, 2024.

Michibata, T. and co-authors: The source of discrepancies in aerosol–cloud–precipitation interactions between GCM and A-Train retrievals, Atmospheric Chemistry and Physics, https://doi.org/10.5194/acp-16-15413-2016, 2016.

Moradi, M.: Wavelet transform approach for denoising and decomposition of satellite-derived ocean color time-series: Selection of optimal mother wavelet, Advances in Space Research, https://doi.org/https://doi.org/10.1016/j.asr.2022.01.023, 2022.

Papacharalampous, G. and co-authors: Predictability of monthly temperature and precipitation using automatic time series forecasting methods, Acta Geophysica, https://doi.org/10.1007/s11600-018-0120-7, 2018.

Quaas, J. and co-authors: Constraining the Twomey effect from satellite observations: issues and perspectives, Atmospheric Chemistry and Physics, https://doi.org/10.5194/acp-20-15079-2020, 2020.

Quan, J. and co-authors: Time series decomposition of remotely sensed land surface temperature and investigation of trends and seasonal variations in surface urban heat islands, Journal of Geophysical Research: Atmospheres, https://doi.org/10.1002/2015JD024354, 2016.

Rabbi, M. F. and Kovács, S.: Quantifying global warming potential variations from greenhouse gas emission sources in forest ecosystems, Carbon Research, https://doi.org/10.1007/s44246-024-00156-7, 2024.

Randles, C. A. and co-authors: Intercomparison of shortwave radiative transfer schemes in global aerosol modeling: results from the AeroCom Radiative Transfer Experiment, Atmospheric Chemistry and Physics, https://doi.org/10.5194/acp-13-2347-2013, 2013.

Recchia, L. G. and Lucarini, V.: Modelling the effect of aerosol and greenhouse gas forcing on the South Asian and East Asian monsoons with an intermediate-complexity climate model, Earth System Dynamics, https://doi.org/10.5194/esd-14-697-2023, 2023.

Stier, P. and co-authors: Host model uncertainties in aerosol radiative forcing estimates: results from the AeroCom Prescribed intercomparison study, Atmospheric Chemistry and Physics, https://doi.org/10.5194/acp-13-3245-2013, 2013.

Stjern, C. W. and co-authors: Systematic Regional Aerosol Perturbations (SyRAP) in Asia Using the Intermediate-Resolution Global Climate Model FORTE2, Journal of Advances in Modeling Earth Systems, https://doi.org/https://doi.org/10.1029/2023MS004171, 2024.

Tang, S. and co-authors: Understanding aerosol–cloud interactions using a single-column model for a cold-air outbreak case during the ACTIVATE campaign, Atmospheric Chemistry and Physics, https://doi.org/10.5194/acp-24-10073-2024, 2024.

van der Werf, G. R. and co-authors: Global fire emissions estimates during 1997–2016, Earth System Science Data, https://doi.org/10.5194/essd-9-697-2017, publisher: Copernicus GmbH, 2017.

van Marle, M. J. E. and co-authors: Historic global biomass burning emissions for CMIP6 (BB4CMIP) based on merging satellite observations with proxies and fire models (1750–2015), Geoscientific Model Development, https://doi.org/10.5194/gmd-10-3329-2017, 2017.

Verbesselt, J. and co-authors: Detecting trend and seasonal changes in satellite image time series, Remote Sensing of Environment, https://doi.org/10.1016/j.rse.2009.08.014, 2010.

Zhang, S. and co-authors: On the contribution of fast and slow responses to precipitation changes caused by aerosol perturbations, Atmos. Chem. Phys., 21, 10179–10197, https://doi.org/10.5194/acp-21-10179-2021, 2021.

Zhou, J. and co-authors: Six-decade temporal change and seasonal decomposition of climate variables in Lake Dianchi watershed (China): stable trend or abrupt shift?, Theoretical and Applied Climatology, https://doi.org/10.1007/s00704-014-1098-y, 2015.